# Socialized Coevolution: Advancing a Better World through Cross-Task Collaboration

**Xinjie Yao** [1 2 3]  **Yu Wang** [1 2 3]  **Pengfei Zhu** [1 2 3]  **Wanyu Lin** [4]  **Ruipu Zhao** [1]  **Zhoupeng Guo** [5]  **Weihao Li** [6]  **Qinghua Hu** [1 2 3]

## Abstract

Traditional machine societies rely on data-driven learning, overlooking interactions and limiting knowledge acquisition from model interplay. To address these issues, we revisit the development of machine societies by drawing inspiration from the evolutionary processes of human societies. Motivated by Social Learning (SL), this paper introduces a practical paradigm of Socialized Coevolution (SC). Compared to most existing methods focused on knowledge distillation and multi-task learning, our work addresses a more challenging problem: not only enhancing the capacity to solve new downstream tasks but also improving the performance of existing tasks through inter-model interactions. Inspired by cognitive science, we propose Dynamic Information Socialized Collaboration (DISC), which achieves SC through interactions between models specialized in different downstream tasks. Specifically, we introduce the dynamic hierarchical collaboration and dynamic selective collaboration modules to enable dynamic and effective interactions among models, allowing them to acquire knowledge from these interactions. Finally, we explore potential future applications of combining SL and SC, discuss open questions, and propose directions for future research, aiming to spark interest in this emerging and exciting interdisciplinary field. Our code will be publicly available at https://github.com/yxjdarren/SC.

[1]College of Intelligence and Computing, Tianjin University, Tianjin, China [2]Engineering Research Center of City Intelligence and Digital Governance, Ministry of Education of the People's Republic of China, Tianjin, China [3]Haihe Lab of ITAI, Tianjin, China [4]Department of Computing, The Hong Kong Polytechnic University, Hong Kong, China [5]School of Automation, Southeast University, Nanjing, China [6]School of New Media and Communication, Tianjin University, Tianjin, China. Correspondence to: Pengfei Zhu <zhupengfei@tju.edu.cn>.

*Proceedings of the $42^{nd}$ International Conference on Machine Learning*, Vancouver, Canada. PMLR 267, 2025. Copyright 2025 by the author(s).

## 1. Introduction

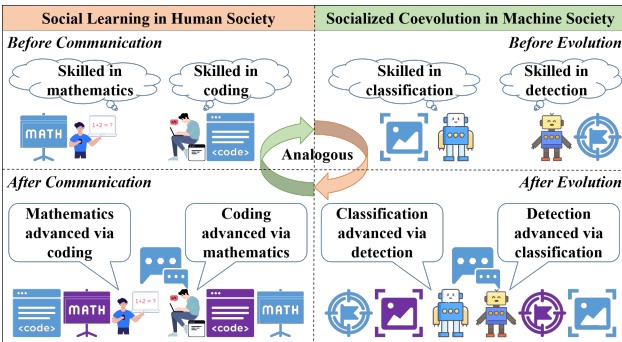

Figure 1: Social learning in human society versus socialized coevolution in machine society.

Similar to human society (Humphrey, 1976; Dunbar, 1998), each model in machine society evolves through knowledge acquisition, as illustrated in Figure 1. Traditional learning paradigms largely depend on data-driven, single-mode learning. While some paradigms (Caruana, 1997; Hinton, 2015; Zhu et al., 2024) allow knowledge transfer from other models, they are limited by blind imitation and difficulty in retaining specialized expertise. In human history, cultural evolution has been driven by continuous capability enhancement, surpassing existing critical skills for survival (Henrich, 2016; Laland, 2017). This enhancement is achieved through ongoing interaction with the environment, rather than extensive trial and data collection. This effective learning paradigm, known as socialized coevolution (SC), has been studied in cognitive science (Thompson et al., 2022).

In machine learning, Knowledge Distillation (KD) (Hinton, 2015) is closely related to SC, as it overcomes the limitation of relying solely on data for knowledge acquisition. However, KD often blindly mimics the teacher model's predictions, which may not always represent valuable knowledge. Unlike humans, KD lacks mutual teaching and learning. An alternative is Multi-Task Learning (MTL) (Caruana, 1997), a shared learning paradigm. While MTL improves performance by leveraging task correlations, it often sacrifices task-specific expertise for generalization, making it hard to maintain performance on individual tasks, particularly when

tasks are highly heterogeneous. Thus, current paradigms are limited in achieving effective coevolution.

Existing paradigms struggle to learn new knowledge and improve capabilities through interaction, as humans do. Due to task and data heterogeneity, models find it difficult to interact effectively. For instance, KD causes blind imitation, hindering the learning of complementary task information and degrading original performance. Similarly, MTL sacrifices task-specific expertise for generalization. In intelligent autonomous systems, the challenge is to acquire more knowledge while maintaining or enhancing original task expertise. By revisiting existing paradigms, we find that two problems are still open: (1) dynamic interactive learning in heterogeneous task scenarios, and (2) expanding general capabilities without compromising task specialization.

To address these problems, we analyze two paradigms: KD and MTL. As shown in Figure 2, both KD and MTL fail to balance the enhancement of original task capabilities and the expansion of general task capabilities due to blind imitation and an overemphasis on generalization. Moreover, KD and MTL rely heavily on large datasets for knowledge transfer, whereas SC enables dynamic cross-task interaction at low cost, facilitating coevolution. Therefore, two significant issues should be explored to realize SC:

1: *How to achieve dynamic interactive learning?*

2: *How to balance specialization and generalization?*

In line with cognitive science (Tomasello et al., 1993; Mesoudi, 2021; Yao et al., 2024), new knowledge should be learned from experts in different tasks and integrated with individual needs for coevolution. To achieve this, we adopt dynamic collaboration, allowing models to selectively interact and acquire valuable knowledge while avoiding blind imitation. Most cultural evolution, driven by population genetics (Mesoudi, 2021) and cognitive science (Tomasello, 2016), inspires our approach, which combines population-genetic-style modeling with directional bias transformation. By organizing experts into a society, maintaining their independence, and enabling interactions, we expand capabilities while enhancing original performance, achieving SC.

To validate this premise, we design an SC-based framework. We first establish that sociability is fundamental to SC, enabling models with complementary strengths to learn mutually. Based on this, the framework is designed around organizational structures, interaction modes, and communication mechanisms. SC is implemented using hierarchical organizational structures, progressive interaction modes, and strong-guided communication mechanisms. Specifically, we create a society of independent classification and detection models, enabling selective collaboration for logits-based mutual learning and hierarchical feature interaction.

These contributions are detailed as follows:

- We introduce a practical learning paradigm, Socialized Coevolution (SC), in which models achieve coevolution through mutual interaction.

- We discuss SC using an information-theoretic framework, where sociability enables models with complementary strengths to mutually and dynamically learn.

- We propose a novel insight into SC methodology, where organizational structures, interaction modes, and communication mechanisms ensure a trade-off between specialization and generalization.

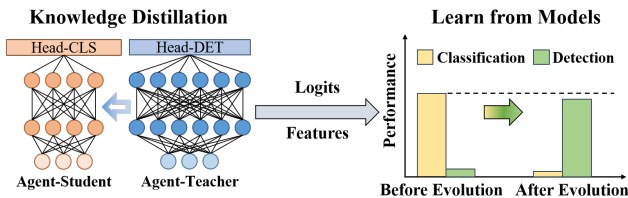

(a) Knowledge distillation paradigm

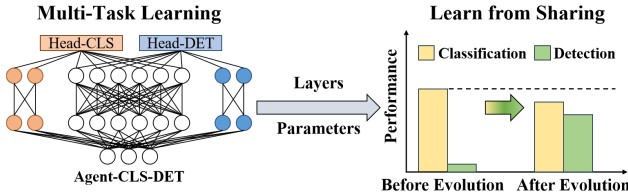

(b) Multi-task learning paradigm

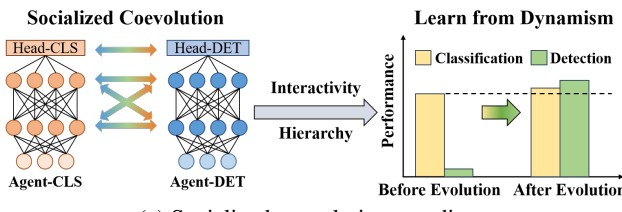

(c) Socialized coevolution paradigm

Figure 2: Comparison of different learning paradigms.

## 2. Related Work

### 2.1. Knowledge Distillation

**Knowledge Distillation (KD)**, as a paradigm for knowledge transfer, focuses on enabling student models to efficiently inherit and apply the knowledge and capabilities of teacher models. Based on the type of information emphasized during the distillation process and the levels at which distillation occurs, existing KD approaches (Gou et al., 2021) can be classified into three categories: **(1) Response-based distillation** emphasizes the outputs or logits of the teacher model, aiming to align the logits of the student model as closely

as possible with those of the teacher model to enhance performance (Jin et al., 2023); (Sun et al., 2024); (Wei et al., 2024). **(2) Feature-based distillation** focuses on knowledge transfer within the intermediate layers of the model, using the feature representations from the teacher model's intermediate layers to guide the corresponding layers of the student model (Hao et al., 2024); (Lu et al., 2024b); (Huang et al., 2024). **(3) Relation-based distillation** targets the relative relationships between different samples in the feature space. By capturing the teacher model's rich representations of these relationships, the student model learns to capture the inherent structure of the data (Zhang et al., 2024); (Yang et al., 2022); (Xiao & Yamasaki, 2024).

Existing KD methods primarily rely on the knowledge obtained by the teacher model, often overlooking the reliability of the knowledge for each individual sample, as illustrated in Figure 2a. The key challenge in KD lies in how to effectively transfer knowledge by considering both the teacher's capabilities and the student's needs. To address this, SC employs dynamic interactive learning to avoid blind reliance, providing insights across different samples.

## 2.2. Multi-Task Learning

**Multi-Task Learning (MTL)** is a paradigm in machine learning that aims to learn multiple related tasks simultaneously, enabling the knowledge from one task to benefit others. This approach seeks to enhance the generalization performance of all tasks involved. Existing methods (Vandenhende et al., 2022) can be broadly categorized into two main groups: **(1) Encoder-focused architectures** share information within the encoder, using either hard- or soft-parameter sharing, before decoding each task with an independent, task-specific head (Agiza et al., 2024); (Xu et al., 2024); (Yang et al., 2024b). **(2) Decoder-focused architectures** first employ a multi-task network to make initial task predictions, and then use features from these initial predictions to further share or exchange information during the decoding stage (Bhattacharjee et al., 2022); (Chen et al., 2024); (Lu et al., 2024a). In addition to the aforementioned approaches, other methods exist outside these categories. For instance, multilinear relationship networks (Long et al., 2017) leverage tensor priors to capture task interactions, while soft layer ordering (Meyerson & Miikkulainen, 2017) enables flexible sharing of network layers across tasks.

Existing MTL research primarily focuses on the sharing of general knowledge across different tasks, often neglecting the preservation of task-specific expertise, as illustrated in Figure 2b. The key challenge in MTL lies in the trade-off between task specialization and generalization. To address this, SC leverages social structures to categorize the model into main and auxiliary decision roles, offering insights into this dilemma.

## 3. Sociability in Coevolution

In this section, we address the issues outlined in Section 1 by providing precise definitions of the problem setup, followed by a concise theoretical analysis. The full proofs of the theorems are provided in the supplementary.

**Problem setups:** Let $\mathcal{M} = \{\mathcal{M}_1, \mathcal{M}_2, \ldots, \mathcal{M}_N\}$ denote a set of $N$ models. Let $X$ represent the input space, and $Y$ denote the label space. The data associated with the $n$-th model is represented as $\mathcal{D}_{\mathcal{M}_n} = \{x_i, y_i\}_{i=1}^{K_{\mathcal{M}_n}}$, where $K_{\mathcal{M}_n}$ denotes the total number of samples for $\mathcal{M}_n$. Each sample $x_i \in \mathbb{R}^D$ corresponds to a label $y_i \in Y_{\mathcal{M}_n}$. To illustrate this setup more clearly, consider two models, $\mathcal{M}_1$ and $\mathcal{M}_2$, as examples. The expert task sets for these two models, $\mathcal{M}_1$ and $\mathcal{M}_2$, are distinct, *i.e.*, $\mathcal{T}_1$ and $\mathcal{T}_2$ represent the sets of tasks at which each model excels. Specifically, the performance of the models on these tasks is described as $\mathcal{M}_1^{\mathcal{T}_1} > \mathcal{M}_2^{\mathcal{T}_1}$ and $\mathcal{M}_1^{\mathcal{T}_2} < \mathcal{M}_2^{\mathcal{T}_2}$, indicating that $\mathcal{M}_1$ outperforms $\mathcal{M}_2$ on task $\mathcal{T}_1$, while $\mathcal{M}_2$ excels on task $\mathcal{T}_2$. After coevolution, the society $\mathcal{S} = \{\mathcal{M}_1, \mathcal{M}_2\}$, formed by the two models, can acquire superior expertise compared to each individual model, *i.e.*, $\mathcal{S}^{\mathcal{T}_1} > \mathcal{M}_1^{\mathcal{T}_1}$ and $\mathcal{S}^{\mathcal{T}_2} > \mathcal{M}_2^{\mathcal{T}_2}$. This collaboration enables broader task coverage and enhances performance on original tasks.

**Definition 3.1.** (Specialization and Generalization) The society $\mathcal{S}$ learns more tasks while improving performance on original tasks, *i.e.*, $\mathcal{T}_S = \mathcal{T}_1 \cup \mathcal{T}_2$, with $\mathcal{S}^{\mathcal{T}_1} > \mathcal{M}_1^{\mathcal{T}_1}$ and $\mathcal{S}^{\mathcal{T}_2} > \mathcal{M}_2^{\mathcal{T}_2}$.

To clearly elucidate the significance of coevolution, we define sociability information and subsequently conduct an analysis based on this premise:

**Definition 3.2.** (Sociability Information) Given the input variables $X_{\mathcal{M}_1}$, $X_{\mathcal{M}_2}$, and the target $Y$, the sociability information provided by the models is defined as:

$$\Phi_{\mathcal{M}_1} = I(X_{\mathcal{M}_1}; Y | X_{\mathcal{M}_2}), \qquad (1)$$

$$\Phi_{\mathcal{M}_2} = I(X_{\mathcal{M}_2}; Y | X_{\mathcal{M}_1}). \qquad (2)$$

The metrics $\Phi_{\mathcal{M}_1}$ and $\Phi_{\mathcal{M}_2}$ quantify the sociability information of $X_{\mathcal{M}_1}$ and $X_{\mathcal{M}_2}$ in SC. Higher values reflect greater sociability and coevolution potential, representing collaboration effectiveness and information exchange between $X_{\mathcal{M}_1}$ and $X_{\mathcal{M}_2}$. The following relation can be derived from information theory:

$$I(X_{\mathcal{M}_1}, X_{\mathcal{M}_2}; Y) = \Phi_{\mathcal{M}_1} + \Phi_{\mathcal{M}_2} + I(X_{\mathcal{M}_1}; X_{\mathcal{M}_2}; Y). \quad (3)$$

**Bayes error rate:** The Bayes error rate (Fukunaga & Hummels, 1987) represents the minimum error, with $P_{e_c}^{mul}$ and $P_{e_c}^{sin}$ denoting the errors for multi-model and single-model scenarios, respectively, for $X_{\mathcal{M}_1}$ and $X_{\mathcal{M}_2}$.

$$P_{e_c}^{mul} = \mathbb{E}_{x_{\mathcal{M}_1}, x_{\mathcal{M}_2} \sim P_{X_{\mathcal{M}_1}, X_{\mathcal{M}_2}}} [1 - \max_{y \in Y} P(Y = y | x_{\mathcal{M}_1}, x_{\mathcal{M}_2})], \quad (4)$$

$$P_{e_c}^{sin} = \mathbb{E}_{x_{\mathcal{M}_1} \sim P_{X_{\mathcal{M}_1}}}[1 - \max_{y \in Y} P(Y = y | x_{\mathcal{M}_1})]. \quad (5)$$

**Theorem 3.3.** *(Key Factor in Coevolution) Building on prior work (Cover, 1999; Feder & Merhav, 1994; Li et al., 2023a), we focus on $P_{e_c}^{mul}$ and $P_{e_c}^{sin}$, as follows:*

$$\frac{H(Y|X_{\mathcal{M}_1}, X_{\mathcal{M}_2}) - \log 2}{\log |Y|} \leq P_{e_c}^{mul} \leq 1 - \exp(-H(Y | X_{\mathcal{M}_1}, X_{\mathcal{M}_2})), \quad (6)$$

$$\frac{H(Y|X_{\mathcal{M}_1}) - \log 2}{\log |Y|} \leq P_{e_c}^{sin} \leq 1 - \exp(-H(Y | X_{\mathcal{M}_1})). \quad (7)$$

Since

$$\Phi_{\mathcal{M}_2} = H(Y|X_{\mathcal{M}_1}) - H(Y|X_{\mathcal{M}_1}, X_{\mathcal{M}_2}). \quad (8)$$

We can derive

$$\frac{H(Y|X_{\mathcal{M}_1}) - \Phi_{\mathcal{M}_2} - \log 2}{\log |Y|} \leq P_{e_c}^{mul} \leq 1 - \exp(-H(Y | X_{\mathcal{M}_1}) + \Phi_{\mathcal{M}_2}). \quad (9)$$

*Remark* 3.4. The difference between $P_{e_c}^{mul}$ and $P_{e_c}^{sin}$ reflects the effect of SC, with $\Phi_{\mathcal{M}_2}$ being the key factor.

**Theorem 3.5.** *(Sociability Information in Data) Building on prior work (Zuo et al., 2024), given the optimal parameter $w^*$ for the data $\mathcal{D}$ and the normalizing constant $p(\mathcal{D})$, we have:*

$$p(\mathcal{D}) = \int p(\mathcal{D} \mid w)p(w)\,dw < \int p(\mathcal{D} \mid w^*)\,p(w)\,dw = p(\mathcal{D} \mid w^*). \quad (10)$$

Since

$$\mathcal{D}_S = \mathcal{D}_{\mathcal{M}_1} \cup \mathcal{D}_{\Phi_{\mathcal{M}_2}}. \quad (11)$$

We can derive

$$\log p(w^* \mid \mathcal{D}_S) > \log p(w^* \mid \mathcal{D}_{\mathcal{M}_1}). \quad (12)$$

*Remark* 3.6. Training on sociability data $\mathcal{D}_{\Phi_{\mathcal{M}_2}}$ enhances $w^*$, proving the effect of SC in optimization.

**Theorem 3.7.** *(Sociability Information in Model) Building on prior work (Zuo et al., 2024), given the optimal parameter $w^*$ consists of the backbone parameter $w_f^*$ and the head parameter $w_h^*$, we have:*

$$\log p\left(w^* \mid \mathcal{D}_{\Phi_{\mathcal{M}_2}}\right) = \log p\left(w_f^* \mid \mathcal{D}_{\Phi_{\mathcal{M}_2}}\right) + \log p\left(w_h^* \mid f\left(\mathcal{D}_{\Phi_{\mathcal{M}_2}}\right)\right). \quad (13)$$

Since

$$\log p\left(w_h^* \mid f\left(\mathcal{D}_{\Phi_{\mathcal{M}_2}}\right)\right) < \log p\left(w_h^* \mid f(\mathcal{D}_{\mathcal{M}_1})^{\Phi_{\mathcal{M}_2}}\right). \quad (14)$$

We can derive

$$\log p\left(w^* \mid \mathcal{D}_{\Phi_{\mathcal{M}_2}}\right) < \log p\left(w_f^* \mid \mathcal{D}_{\Phi_{\mathcal{M}_2}}\right) + \log p\left(w_h^* \mid f(\mathcal{D}_{\mathcal{M}_1})^{\Phi_{\mathcal{M}_2}}\right). \quad (15)$$

*Remark* 3.8. Training on sociability features $f(\mathcal{D}_{\mathcal{M}_1})^{\Phi_{\mathcal{M}_2}}$ enhances $w^*$, proving the effect of SC in optimization.

# 4. Methodology

In this section, to address the two issues highlighted in Section 1, we introduce dynamic interactive learning as a means to balance specialization and generalization. This approach is realized in two ways. First, we facilitate dynamic interactions between teacher and student models, allowing them to adjust based on their respective strengths and needs. Second, we organize the models into a society where, depending on the task, some models take on the role of decision-makers while others assist in the main process. The concepts of mutual growth through teaching and leveraging individual strengths are considered forms of SC.

We first analyze the driving factors of SC, introduce its core framework, and then detail the dynamic interactive learning mechanism along with our approach to balancing specialization and generalization.

## 4.1. Key Factors of SC

In the face of the challenge to balance specialization and generalization within SC, a natural question arises: what are the key factors influencing SC?

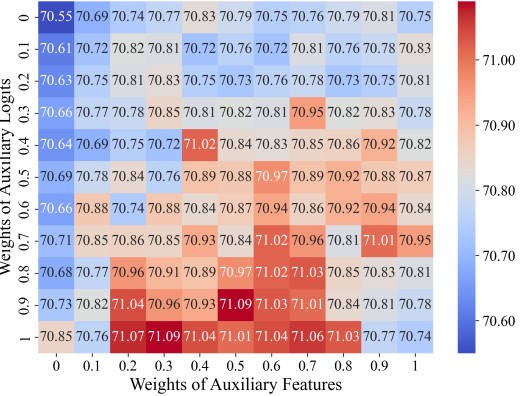

(a) Comparison of different weights on CIFAR100

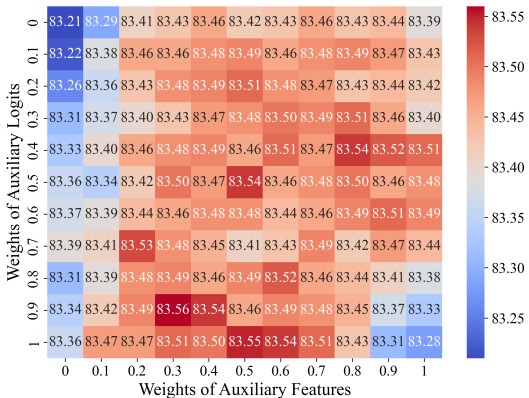

(b) Comparison of different weights on VOC07+12

Figure 3: Comparison of different weights on datasets.

To address the aforementioned question, we consider two downstream tasks: classification and detection. Specifically, features provide patch-wise information, while logits offer image-wise information. Consequently, we conduct exploratory experiments on the interaction between features and logits under varying weights. From Figure 3, it can be inferred that neither extremely low nor high weights are suitable for effective interaction. When the weight is too low, the knowledge interaction provides limited assistance, failing to improve the performance. Conversely, when the weight is too high, the knowledge becomes overly complex for the model to learn effectively. Notably, the model exhibits superior performance under a variety of weight combinations. Based on this observation, we hypothesize that a **dynamic weighting** mechanism for interaction may effectively balance specialization and generalization.

**Theorem 4.1.** *(Generalization Bound of Collaboration) Building on prior work (Zhang et al., 2023), we focus on the dynamic weighting mechanism. Let $D_{train} = \{x_i, y_i\}_{i=1}^N$ be a training dataset of $N$ samples, and $e\hat{r}r(f^m)$ be the empirical error of the $m$-th model $f^m$ on $D_{train}$. For any hypothesis $f \in \mathcal{H}$ (i.e., $\mathcal{H} : \mathcal{X} \to \{-1, 1\}$), with probability at least $1 - \delta$, the generalization error is bounded by:*

$$
\begin{aligned}
GError(f) \leq & \underbrace{\sum_{m=1}^{M} \mathbb{E}(w^m) e\hat{r}r(f^m)}_{\text{Term-L (empirical loss)}} + \underbrace{\sum_{m=1}^{M} \mathbb{E}(w^m)\mathfrak{R}_m(f^m)}_{\text{Term-C (complexity)}} \\
& + \underbrace{\sum_{m=1}^{M} Cov(w^m, \ell^m)}_{\text{Term-Cov (covariance)}} + M\sqrt{\frac{\ln(1/\delta)}{2N}},
\end{aligned}
$$
(16)

where $\mathbb{E}(w^m)$ is the expected collaboration weights, $\mathfrak{R}_m(f^m)$ is the Rademacher complexity of model $f^m$, and $Cov(w^m, \ell^m)$ is the covariance between the collaborative weight and the loss. In static collaboration, the weights $w_{static}^m$ remain constant, so $Cov(w_{static}^m, \ell^m) = 0$. By contrast, in dynamic collaboration, the weight $w_{dynamic}^m$ increases as the loss $\ell^m$ decreases, yielding $Cov(w_{dynamic}^m, \ell^m) \leq 0$. This negative covariance lowers Term-Cov, thereby tightening the upper bound of generalization error of collaboration, *i.e.*, $\mathcal{O}(GError(f_{dynamic})) \leq \mathcal{O}(GError(f_{static}))$.

*Remark* 4.2. Dynamic collaboration yields a tighter generalization error bound compared to static collaboration.

## 4.2. Socialized Coevolution Framework

We take a closer look at SC to identify the key factors forming its framework. Inspired by (Zuo et al., 2024; Li et al., 2023a), we perform an information-theoretic analysis of SC in the previous section and explore the impact of sociability information. Based on this, we focus on three

aspects: organizational structures, interaction modes, and communication mechanisms, aiming to design a unified SC framework $U(\cdot)$ and implement dynamic interactive learning to balance specialization and generalization.

$$
U(\mathcal{M}_1, \mathcal{M}_2) = \psi(\varphi(\mathcal{M}_1, \mathcal{M}_2), \beta), \quad (17)
$$

where $\mathcal{M}_1$ and $\mathcal{M}_2$ are two models, $\varphi(\cdot)$ represents the organizational structures, $\beta$ denotes the communication mechanisms, and $\psi(\cdot)$ represents the interaction modes.

### 4.3. Dynamic Information Socialized Collaboration

Driven by the above analysis and (Li et al., 2023b; Shao et al., 2021; Cao et al., 2024), our goal is to address SC from three aspects: hierarchical organizational structures, progressive interaction modes, and strong-guided communication mechanisms. We introduce a new approach based on SC, called Dynamic Information Socialized Collaboration (DISC), as shown in Figure 4. For clarity, we provide a detailed explanation of the three aspects included in DISC.

**Hierarchical organizational structures:** In DISC, we design the Dynamic Hierarchical Collaboration (DHC) module to effectively leverage hierarchical auxiliary information, aiding the main model in learning. Specifically, for different tasks, we treat the specialized model as the main entity and the other models as auxiliary information. For instance, in classification tasks, the model specialized in classification focuses on global-wise information, while the model specialized in detection emphasizes patch-wise information. The key to achieving SC lies in fully utilizing the patch-wise knowledge embedded in the detection model. The hierarchical organizational structure facilitates the progressive abstraction and capture of both low-level and high-level image features, enabling effective handling of complex visual tasks. Therefore, we define patch-wise information in the form of a hierarchical structure, as follows:

$$
input_{main}^l = output_{main}^{l-1} + DHC(l-1), \quad (18)
$$

where $input_{main}^l$ denotes the input of the $l$-th layer of the main model, $output_{main}^{l-1}$ denotes the output of the $(l-1)$-th layer of the main model, and $DHC(l-1)$ denotes the hierarchical output of the auxiliary model. Next, we will provide a detailed explanation of $DHC(l-1)$ in progressive interaction modes.

**Progressive interaction modes:** In DISC, DHC mitigates excessive reliance on auxiliary information by employing progressive interaction, effectively utilizing hierarchical auxiliary data to support the main model's learning process. Initially, the model depends heavily on data, but as optimization advances, the information provided by the data becomes limited. At this stage, gradually incorporating auxiliary information from other dimensions becomes essential for further learning. Overuse of auxiliary information early

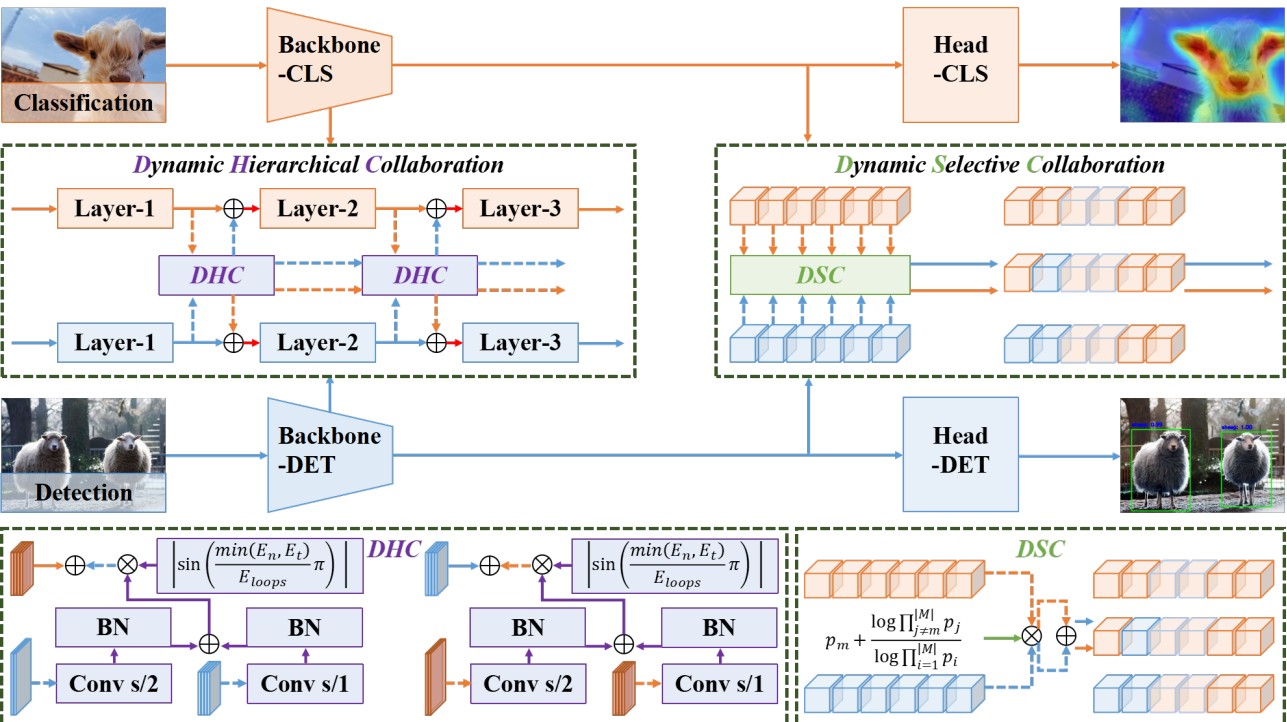

Figure 4: An overview of our proposed Dynamic Information Socialized Collaboration (DISC). We design the Dynamic Hierarchical Collaboration (DHC) and Dynamic Selective Collaboration (DSC) modules, integrating organizational structures, interaction modes, and communication mechanisms as the core elements of SC.

in the process can impede the model's ability to learn from data. Thus, the key to SC lies in the gradual integration of auxiliary information, enabling the model to overcome the limitations of data-driven learning. This progressive interaction mode is defined through DHC as follows:

$$DHC\,(l-1) = \left| \sin\left( \frac{\min(E_n, E_t)}{E_{loops}}\pi \right) \right| \sum_{i=1}^{l-1} output_{aux}^i, \quad (19)$$

where $E_n$ represents the current training epoch, $E_t$ denotes the threshold epoch, $E_{loops}$ indicates the total number of training epochs, and $output_{aux}^i$ denotes the output of the $i$-th layer of the auxiliary model.

**Strong-guided communication mechanisms:** In DISC, we designed the Dynamic Selective Collaboration (DSC) module to leverage strong-guided communication mechanisms to enhance the learning process of the main model. Specifically, the weights assigned to the main and auxiliary models vary dynamically depending on the sample, meaning the weight of the main model is not necessarily greater than that of the auxiliary model. Fixing the weights as constants prevents the models from adapting flexibly to their performance. The key to achieving SC is to assign appropriate weights dynamically to each model. A model's weight should negatively correlate with its own loss and positively correlate with the losses of other models. Strong-guided communication mechanisms dynamically assign weights

based on how well each model handles different samples, enabling effective learning for every sample. This dynamic weighting is defined through strong-guided communication mechanisms as follows:

$$DSC\,(m) = \left( p_m + \frac{\log \prod_{j \neq m}^{|\mathcal{M}|} p_j}{\log \prod_{i=1}^{|\mathcal{M}|} p_i} \right) \cdot output_m^{logits}, \quad (20)$$

where $p_m$ is the prediction of $m$-th model.

The DHC and DSC modules, featuring hierarchical organizational structures, progressive interaction modes, and strong-guided communication mechanisms, are integrated into the overall loss function as follows:

$$\begin{aligned} \mathcal{L}_{overall} =& \mathcal{L}_{cls}\left( y, f^{DSC}(f^{DHC}(x)) \right) \\ &+ \mathcal{L}_{reg}\left( y, f^{DSC}(f^{DHC}(x)) \right), \end{aligned} \quad (21)$$

where $x$ is the input, $y$ is the label, $\mathcal{L}_{cls}$ represents the classification loss, $\mathcal{L}_{reg}$ represents the regression loss, $f^{DHC}(\cdot)$ denotes the collaboration with DHC, and $f^{DSC}(\cdot)$ denotes the collaboration with DSC. $\mathcal{L}_{cls}$ is employed for classification tasks, while both $\mathcal{L}_{cls}$ and $\mathcal{L}_{reg}$ are utilized for detection tasks. For a clearer understanding of training and inference, we have described the detailed algorithms in Algorithms 1 and 2.

**Algorithm 1** Training for DISC.

**Input**: Datasets $\mathcal{D}_{\mathcal{M}_n}$, model $\mathcal{M}_n$, main task $\mathcal{T}_{main}$;
**Output**: Embedding with DHC $f_{\mathcal{M}_n}^{DHC}(\cdot)$, embedding with DSC $f_{\mathcal{M}_n}^{DSC}(\cdot)$, downstream task head $f_{\mathcal{M}_n}^{head}(\cdot)$;

1: **if** $\mathcal{T}_{main}$ = classification **then**
2:    **while** not converged **do**
3:       Fix detection model $\mathcal{M}_{det}$;
4:       Get a mini-batch of training data from $\mathcal{D}_{\mathcal{M}_{cls}}$;
5:       Calculate the overall loss $\mathcal{L}_{overall}$ using Eq. (21);
6:       Update the classification model, *i.e.*, $f_{\mathcal{M}_{cls}}^{DHC}(\cdot)$, $f_{\mathcal{M}_{cls}}^{DSC}(\cdot)$, and $f_{\mathcal{M}_{cls}}^{head}(\cdot)$;
7:    **end while**
8: **end if**
9: **if** $\mathcal{T}_{main}$ = detection **then**
10:   **while** not converged **do**
11:      Fix classification model $\mathcal{M}_{cls}$;
12:      Get a mini-batch of training data from $\mathcal{D}_{\mathcal{M}_{det}}$;
13:      Calculate the overall loss $\mathcal{L}_{overall}$ using Eq. (21);
14:      Update the detection model, *i.e.*, $f_{\mathcal{M}_{det}}^{DHC}(\cdot)$, $f_{\mathcal{M}_{det}}^{DSC}(\cdot)$, and $f_{\mathcal{M}_{det}}^{head}(\cdot)$;
15:   **end while**
16: **end if**

---

**Algorithm 2** Inference for DISC.

**Given components:** Backbone $f_{\mathcal{M}_n}^b(\cdot)$, downstream task head $f_{\mathcal{M}_n}^{head}(\cdot)$;
**Input**: Test sample $x$;
**Output**: Final prediction $y^*$;

1: Calculate image feature $f_{\mathcal{M}_{cls}}^b(x)$ and $f_{\mathcal{M}_{det}}^b(x)$;
2: Calculate weights of the feature $\mathcal{W}_{cls}^{DHC}$ and $\mathcal{W}_{det}^{DHC}$;
3: Calculate weights of the logits $\mathcal{W}_{cls}^{DSC}$ and $\mathcal{W}_{det}^{DSC}$;
4: Return final prediction $y^*$.

# 5. Experiments

In this section, we compare DISC with state-of-the-art methods on the CIFAR100 and VOC07+12 datasets. We discuss task-driven knowledge transfer methods, and the ablation studies demonstrate the effectiveness of the DHC and DSC modules. Our model is implemented in PyTorch (Paszke et al., 2019) and deployed on an NVIDIA RTX 3090 GPU.

## 5.1. Implementation Details

**Datasets:** We evaluate the proposed method on CIFAR100 (Krizhevsky et al., 2009) and VOC07+12 (Everingham et al., 2010; 2015) datasets using a data-efficient setting. Specifically, we employ 10% of the training set and the complete test set, with CIFAR100 used for evaluating classification and VOC07+12 for assessing detection.

**Compared methods:** To enable comparison with KD and MTL in the SC setting, we adopt the hard-sharing mecha-

nism from MTL, based on KD methods such as LSKD (Sun et al., 2024), CrossKD (Wang et al., 2024), and PPAL (Yang et al., 2024a). In this approach, the hidden layers trained under the guidance of the teacher model are shared across all tasks, while separate output layers for specific tasks are retained. Additionally, we compare several classic methods, including SwAV (Caron et al., 2020), DeepClusterV2 (Caron et al., 2018), MoCo v2 (Chen et al., 2020), and CLIP (Radford et al., 2021).

## 5.2. Specialization and Generalization are All You Need

We analyze the Fine-Tune (FT), KD, and MTL knowledge transfer methods based on expert models for classification and detection, as shown in Table 1. Then, we compare DISC with state-of-the-art methods, as shown in Table 2.

| Method | Before evolution (%) ↑ | | | After evolution (%) ↑ | | |
|---|---|---|---|---|---|---|
| | CLS | DET | AVG | CLS | DET | AVG |
| Expert(CLS)+FT | 70.68 | 0.00 | 35.34 | 0.68 | 65.00 | 32.84 |
| Expert(CLS)+KD | 70.68 | 0.00 | 35.34 | 0.61 | 66.35 | 33.48 |
| Expert(CLS)+MTL | 70.68 | 0.00 | 35.34 | **70.68** | 59.92 | **65.30** |
| Expert(DET)+FT | 0.47 | 83.89 | 42.18 | 64.08 | 0.00 | 32.04 |
| Expert(DET)+KD | 0.47 | 83.89 | 42.18 | 66.82 | 0.00 | 33.41 |
| Expert(DET)+MTL | 0.47 | 83.89 | 42.18 | 43.05 | **83.89** | 63.47 |
| DISC | 70.68 | 0.00 | 35.34 | **72.28**$_{(+1.60)}$ | **84.92**$_{(+1.03)}$ | **78.60**$_{(+13.30)}$ |

Table 1: Comparison of performance across different tasks before and after evolution on CIFAR100 and VOC07+12 datasets. The $1^{st}/2^{nd}$ best results are indicated in red/blue.

| Method | CLS (%) ↑ | DET (%) ↑ | AVG (%) ↑ |
|---|---|---|---|
| SwAV (Caron et al., 2020) | 65.90 | 69.20 | 67.55 |
| DeepClusterV2 (Caron et al., 2018) | 66.10 | 70.00 | 68.05 |
| MoCo v2 (Chen et al., 2020) | 66.00 | 70.20 | 68.10 |
| CLIP (Radford et al., 2021) | 58.70 | 68.60 | 63.65 |
| LSKD(CLS)+MTL (Sun et al., 2024) | 71.05 | 63.37 | 67.21 |
| LSKD(DET)+MTL (Sun et al., 2024) | 50.13 | 83.91 | 67.02 |
| CrossKD(CLS)+MTL (Wang et al., 2024) | **71.13** | 63.85 | 67.49 |
| CrossKD(DET)+MTL (Wang et al., 2024) | 52.96 | **84.01** | **68.49** |
| PPAL(CLS)+MTL (Yang et al., 2024a) | 70.95 | 62.31 | 66.63 |
| PPAL(DET)+MTL (Yang et al., 2024a) | 50.68 | 83.95 | 67.32 |
| DISC | **72.28**$_{(+1.15)}$ | **84.92**$_{(+0.91)}$ | **78.60**$_{(+10.11)}$ |

Table 2: Comparison of performance across different tasks on CIFAR100 and VOC07+12 datasets. The $1^{st}/2^{nd}$ best results are indicated in red/blue.

**FT and KD lose task generalization:** As shown in Table 1, both FT and KD obtain new downstream task capabilities at the cost of sacrificing performance on existing tasks. This raises the question of whether acquiring new task abilities alone constitutes evolution. Our goal is for the model to retain or even improve performance on old tasks while learning new ones. Additionally, even focusing on new tasks, FT and KD underperform compared to DISC, as they fail to utilize cross-task knowledge. Both FT and KD can cause models to blindly follow the evolutionary process, degrading original task capabilities and losing generalization.

**MTL overlooks specialization for new tasks:** As shown in Table 1, we observe that MTL preserves performance on existing tasks by freezing the model for those tasks. While MTL trains new task-specific parameters for learning new downstream tasks, the model cannot fully optimize for the new tasks due to the need to maintain performance on existing ones. MTL achieves basic evolution but still falls short of our goal. MTL struggles to fully learn the capabilities for new downstream tasks while failing to enhance performance on existing tasks. In the evolutionary process, MTL prioritizes maintaining generalization for existing tasks, overlooking specialization for new tasks.

**DISC balances specialization and generalization:** Based on the SC paradigm, DISC trains task-specific parameters while dynamically leveraging auxiliary information from other models for collaborative learning. DISC's hierarchical organizational structures utilize both task-specific data and guidance from other tasks, acquiring intra-task and cross-task knowledge. Its progressive interaction modes enable curriculum-like learning, while strong-guided communication mechanisms allow dynamic weighting of learning and inference. DISC not only enhances new task capabilities but also improves existing task performance through model interactions, as shown in Tables 1 and 2.

**Analysis of specialization and generalization:** Specialization and generalization are critical in SC. Specialization means the model can effectively learn new tasks or improve performance on existing ones, while generalization refers to the model's ability to perform well across multiple tasks. As shown in Table 2, classical models struggle to maintain their original performance in data-efficient settings and on ResNet50, indicating an over-reliance on large datasets and model parameters. Although the combination of KD and MTL comes close to achieving the SC objective, there is still a noticeable gap compared to the DISC model we propose.

### 5.3. Ablation Study

To further verify the significance of each module, *i.e.*, DHC and DSC, in DISC based on SC, we conduct the ablation study as shown in Table 3. The results reveal that the model with DHC and DSC attains the best performance.

| Method | DHC | DSC | CLS (%) ↑ | DET (%) ↑ | AVG (%) ↑ |
|--------|-----|-----|-----------|-----------|-----------|
|        |     |     | 70.74     | 83.28     | 77.01     |
| DISC   | ✓   |     | 71.26     | 84.25     | 77.76     |
|        |     | ✓   | 71.15     | 84.04     | 77.60     |
|        | ✓   | ✓   | **72.28**$_{(+1.54)}$ | **84.92**$_{(+1.64)}$ | **78.60**$_{(+1.59)}$ |

Table 3: Ablation study on CIFAR100 and VOC07+12. In the table, "✓" denotes DISC with the module.

**Without DHC and DSC:** We observe that DHC and DSC are essential to DISC, as they effectively utilize the socia-bility information of cross-task models. In contrast, direct interaction between different downstream task models follows a static interaction mechanism that aggregates features from the same layer with uniform weights, which restricts improvements in both specialization and generalization.

**Using only DHC:** We observe significant performance improvement. This indicates that hierarchical cross-task features interacting dynamically and progressively are highly effective in enhancing representational capability. DHC provides insight into cross-task knowledge transfer.

**Using only DSC:** We observe that the model's performance improves, but not as significantly as with DHC. While DHC focuses on enhancing representational capability, DSC focuses on improving decision-making capability. This suggests that both representational and decision-making capabilities are crucial for model performance. DSC provides insight into dynamic decision-making.

**Combining DHC and DSC:** We find that the model achieves the expected results, not only adding new downstream task capabilities but also improving the performance of existing tasks through model interactions. This shows that representational ability forms the foundation for achieving good performance, while decision-making ability helps push the performance beyond its limits.

### 5.4. Further Analysis

The DISC, designed with SC, incorporates task-specific characteristics through the DHC and DSC modules. Classification focuses on image-wise logits interaction, while detection emphasizes patch-wise feature interaction. Despite their different focuses, the complementary information from both tasks enhances representational and decision-making capabilities, as illustrated in Figure 5.

**Adversarial competition:** As shown in the red dashed box in Figure 5a, DHC and DSC do not consistently improve performance across all classes and may even experience performance degradation due to adversarial competition. DHC focuses on low-level intra-class information, while DSC emphasizes high-level inter-class information. Relying on only one of these modules is affected by large intra-class variation and high inter-class similarity, leading to adversarial competition.

**Dynamic coevolution:** As illustrated in Figure 5b, the synergistic integration of DHC and DSC significantly improves class-wise performance, culminating in enhanced overall model efficacy. This underscores the critical role of both low-level and high-level information in augmenting the model's perceptual capabilities. Mastering the effective collaboration of this complementary information is fundamental to realizing SC, with DISC offering valuable insights into this complex interplay.

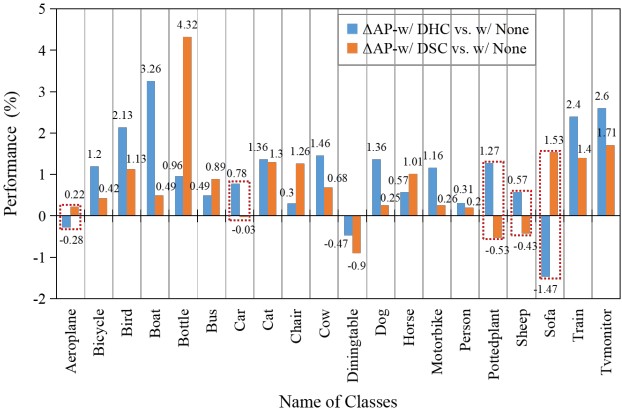

(a) Comparison of various modules on VOC07+12 dataset

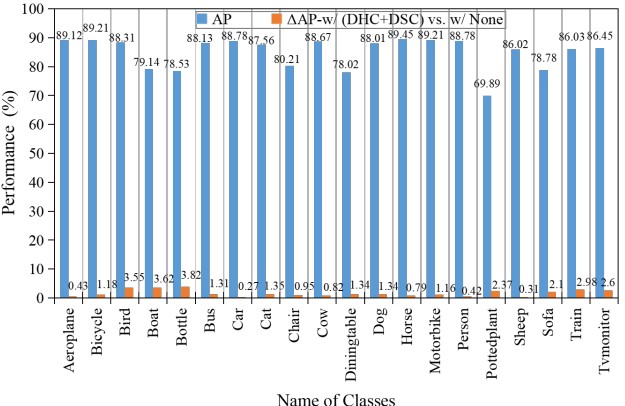

(b) Comparison of DISC and vanilla on VOC07+12 dataset

Figure 5: The performance differences for each class.

## 5.5. Discussion

SC is a cross-task collaborative paradigm that facilitates knowledge interaction, allowing models to co-evolve while improving both specialization and generalization. Alongside advancing learning paradigms, SC is applicable to defect detection, autonomous driving, and emergency rescue. Table 4 presents a comparative analysis of learning paradigms in addressing diverse challenges.

| Paradigm | Specialization | Generalization | Dynamism | Hierarchy |
|----------|:---:|:---:|:---:|:---:|
| KD | ✓ | ✗ | ✗ | ✗ |
| MTL | ✗ | ✓ | ✗ | ✗ |
| SC | ✓ | ✓ | ✓ | ✓ |

Table 4: Comparison of different learning paradigms.

**i) Specialization:** The model needs to preserve or improve the performance of original tasks while learning new tasks.

**ii) Generalization:** As new tasks arise, the model needs to adapt without compromising existing performance.

**iii) Dynamism:** Model interaction should be weighted by each model's proficiency with each sample.

**iv) Hierarchy:** Model interaction should be diverse and comprehensive to maximize knowledge utilization.

Existing KD and MTL paradigms face challenges in balancing specialization and generalization in dynamic environments. SC overcomes this by enabling knowledge sharing and coevolution via DHC and DSC modules across different downstream tasks.

## 6. Conclusion

In this paper, we introduce a practical SC paradigm with a rigorous mathematical foundation and an information-theoretic explanation. Additionally, we carefully design DISC based on SC to not only incorporate new downstream task capabilities but also enhance the performance of existing tasks through model interactions. Essentially, we view specialization and generalization as two sides of the same coin, not independent issues. The dynamic interaction between features and logits provides new insights into SC, highlighting the importance of tailored approaches to maximize strengths. In future work, we plan to further explore the potential of combining multiple modalities and tasks.

## Acknowledgements

This work was supported in part by the National Science and Technology Major Project under Grant 2022ZD0116500, in part by the National Natural Science Foundation of China under Grants 62436002, 62476195, U23B2049, and 62222608, in part by Tianjin Natural Science Funds for Distinguished Young Scholar under Grant 23JCJQJC00270, in part by Zhejiang Provincial Natural Science Foundation of China under Grant LD24F020004, in part by Tianjin Young Scientific and Technological Talents Project under grant QN20230305, in part by Tianjin Science and Technology Plan Project under grant 24YDTPJC00150, in part by Natural Science Foundation of Tianjin under grant 24JCY-BJC00950, and in part by the Huawei Ascend Computing Platform.

## Impact Statement

This paper presents work aimed at advancing the field of coevolution in machine learning. Our goal is to propose a socialized coevolution learning paradigm to enhance the performance of existing downstream tasks and facilitate the learning of new downstream tasks, making full use of the complementarities between different tasks. However, due to the presence of model heterogeneity in dynamic open environments, the application of our approach to real-world scenarios may inevitably encounter heterogeneity challenges.

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

# A. Supplementary Material

The supplementary material contains comprehensive details on the implementations and experimental results referenced in the main paper, along with supplementary theoretical analysis and in-depth discussions. It is organized as follows:

- In Section B, we offer a thorough overview of the methods compared in the main paper, accompanied by a detailed description of the datasets utilized.

- In Section C, we present the full set of experimental results, accompanied by an in-depth analysis that thoroughly evaluates the model's performance.

- In Section D, we provide a comprehensive theoretical analysis of sociability information, exploring its implications within the context of the study.

- In Section E, we explore SC's insights across real-world applications and learning paradigms, aiming to spark interest in this dynamic interdisciplinary field.

# B. Implementation Details

In this section, we offer a detailed description of the methods compared in the main paper, as well as the datasets used. For classification as the primary task, the model is trained for 1500 epochs using SGD with a batch size of 64. The threshold is set to 300, with a learning rate of 0.01, momentum of 0.9, and weight decay of 0.001. For detection as the primary task, the model is trained for 600 epochs using SGD with a batch size of 32. The threshold is also set to 300, with a learning rate of 0.03, momentum of 0.9, and weight decay of 0.001. All reported results represent the mean values obtained from 3 independent trials.

## B.1. Compared Methods

In this subsection, we provide an overview of the methods compared in the main paper, outlining their key characteristics. The methods considered are as follows:

- **SwAV (Caron et al., 2020):** state-of-the-art unsupervised learning method, which improves representation by using online clustering with swapped prediction.

- **DeepClusterV2 (Caron et al., 2018):** state-of-the-art unsupervised learning method, which jointly trains convolutional networks and uses k-means clustering for end-to-end optimization.

- **MoCov2 (Chen et al., 2020):** state-of-the-art contrastive learning method, which improves image representation by incorporating a projection head and stronger data augmentation.

- **CLIP (Radford et al., 2021):** state-of-the-art contrastive learning method, which pre-trains on a large dataset of pairs and enables zero-shot transfer across various tasks without task-specific training.

- **LSKD (Sun et al., 2024):** state-of-the-art knowledge distillation method, which standardizes logits using Z-score before softmax.

- **CrossKD (Wang et al., 2024):** state-of-the-art knowledge distillation method, which aligns student and teacher outputs through cross-head mimicking.

- **PPAL (Yang et al., 2024a):** state-of-the-art active learning method, which combines uncertainty-based and diversity-based sampling.

## B.2. Datasets

The VOC07+12 training dataset contains 47,221 objects, while the testing dataset includes 14,976 objects. The CIFAR100 training dataset consists of 50,000 images, with 10,000 images in the testing set. In the data-efficient setting, we utilize 10% of the training data: for VOC07+12 (10%), the training set contains 4,696 objects, and the testing set remains at 14,976 objects. For CIFAR100 (10%), the training set includes 5,000 images, with the testing set consisting of 10,000 images. We list the details of the datasets in Table 5.

# C. Full Experimental Results

In this section, we present additional experimental results, including analyses of parameters and visualizations.

## C.1. Analysis of Parameters

**Dynamic progressive interaction:** Inspired by curriculum learning (Bengio et al., 2009), we apply sine and cosine in Dynamic Information Socialized Collaboration (DISC) to achieve dynamic progressive interaction. DISC incorporates two key hyper-parameters: training loops $E_{\text{loops}}$ and threshold $E_t$, both of which influence the dynamic progressive interaction of the model. As shown in Figure 6, the sine function achieves superior performance on CIFAR100 when $E_{\text{loops}} = 1500$ and $E_t = 300$, while $E_{\text{loops}} = 600$ and $E_t = 300$ yield better results on VOC07+12.

## C.2. Analysis of Visualizations

**Competition and coevolution:** In the main paper, we analyze Figure 5 to demonstrate that using Dynamic Hierarchical Collaboration (DHC) or Dynamic Selective Collaboration (DSC) alone cannot fully realize coevolution for certain categories, due to high intra-class variance and inter-class similarity. Only their combined application effectively overcomes these challenges. To further investigate, we perform

| Datasets | Training dataset | Testing dataset | Detailed classes |
| --- | --- | --- | --- |
| VOC07+12 | 47,221 | 14,976 | aeroplane, bicycle, bird, boat, bus, car, cat, chair, cow, dining table, dog, horse, motorbike, person, potted plant, sheep, sofa, train, TV monitor |
| VOC07+12(10%) | 4,696 | 14,976 | aeroplane, bicycle, bird, boat, bus, car, cat, chair, cow, dining table, dog, horse, motorbike, person, potted plant, sheep, sofa, train, TV monitor |
| CIFAR100 | 50,000 | 10,000 | apple, aquarium fish, baby, bear, beaver, bed, bee, beetle, bicycle, bottle, bowl, boy, bridge, bus, butterfly, camel, can, castle, caterpillar, cattle, chair, chimpanzee, clock, cloud, cockroach, couch, crab, crocodile, cup, dinosaur, dolphin, elephant, flatfish, forest, fox, girl, hamster, house, kangaroo, keyboard, lamp, lawn mower, leopard, lion, lizard, lobster, man, maple tree, motorcycle, mountain, mouse, mushroom, oak tree, orange, orchid, otter, palm tree, pear, pickup truck, pine tree, plain, plate, poppy, porcupine, possum, rabbit, raccoon, ray, road, rocket,rose, sea, seal, shark, shrew, skunk, skyscraper, snail, snake, spider, squirrel, streetcar, sunflower, sweet pepper, table, tank, telephone, television, tiger, tractor, train, trout, tulip, turtle, wardrobe, whale, willow tree, wolf, woman, worm |
| CIFAR100(10%) | 5,000 | 10,000 | apple, aquarium fish, baby, bear, beaver, bed, bee, beetle, bicycle, bottle, bowl, boy, bridge, bus, butterfly, camel, can, castle, caterpillar, cattle, chair, chimpanzee, clock, cloud, cockroach, couch, crab, crocodile, cup, dinosaur, dolphin, elephant, flatfish, forest, fox, girl, hamster, house, kangaroo, keyboard, lamp, lawn mower, leopard, lion, lizard, lobster, man, maple tree, motorcycle, mountain, mouse, mushroom, oak tree, orange, orchid, otter, palm tree, pear, pickup truck, pine tree, plain, plate, poppy, porcupine, possum, rabbit, raccoon, ray, road, rocket,rose, sea, seal, shark, shrew, skunk, skyscraper, snail, snake, spider, squirrel, streetcar, sunflower, sweet pepper, table, tank, telephone, television, tiger, tractor, train, trout, tulip, turtle, wardrobe, whale, willow tree, wolf, woman, worm |

Table 5: Detailed datasets.

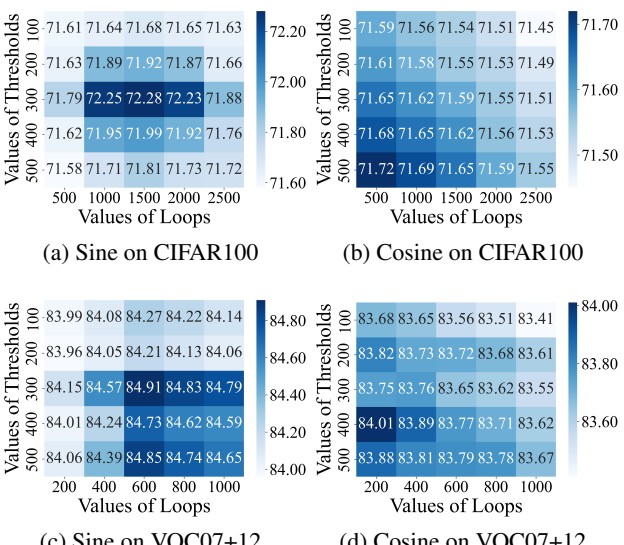

(a) Sine on CIFAR100     (b) Cosine on CIFAR100

(c) Sine on VOC07+12     (d) Cosine on VOC07+12

Figure 6: Comparison of different weights.

Class Activation Maps (CAMs) analysis (Zhou et al., 2016), as shown in Figure 7. DHC emphasizes low-level intra-class features, which can cause confusion for categories with sim-ilar textures, while DSC focuses on high-level inter-class features, which may confuse categories with high inter-class similarity. To address these issues, we integrate DHC and DSC into the DISC model, eliminating adversarial competition and promoting coevolution.

## D. Theoretical Analysis

Below, we provide the omitted proofs from Sections 3 and 4 (refer to the main paper).

### D.1. Proof of Theorem 3.3

*Proof.* We leverage previous results (Cover, 1999; Feder & Merhav, 1994; Li et al., 2023a):

$$-\log\left(1 - P_{e_c}^{mul}\right) \leq H(Y \mid X_{\mathcal{M}_1}, X_{\mathcal{M}_2}), \quad (22)$$

$$H(Y \mid X_{\mathcal{M}_1}, X_{\mathcal{M}_2}) \leq \log 2 + P_{e_c}^{mul} \log |Y|. \quad (23)$$

Combine the two inequalities and put $P_{e_c}^{mul}$ in the middle:

$$\frac{H(Y|X_{\mathcal{M}_1}, X_{\mathcal{M}_2}) - \log 2}{\log |Y|} \leq P_{e_c}^{mul} \leq 1 - \exp(-H(Y \mid X_{\mathcal{M}_1}, X_{\mathcal{M}_2})), \quad (24)$$

which is the first result in the theorem. Then we apply the results to $P_{e_c}^{sin}$:

$$\frac{H(Y|X_{\mathcal{M}_1}) - \log 2}{\log |Y|} \leq P_{e_c}^{sin} \leq 1 - \exp(-H(Y \mid X_{\mathcal{M}_1})). \quad (25)$$

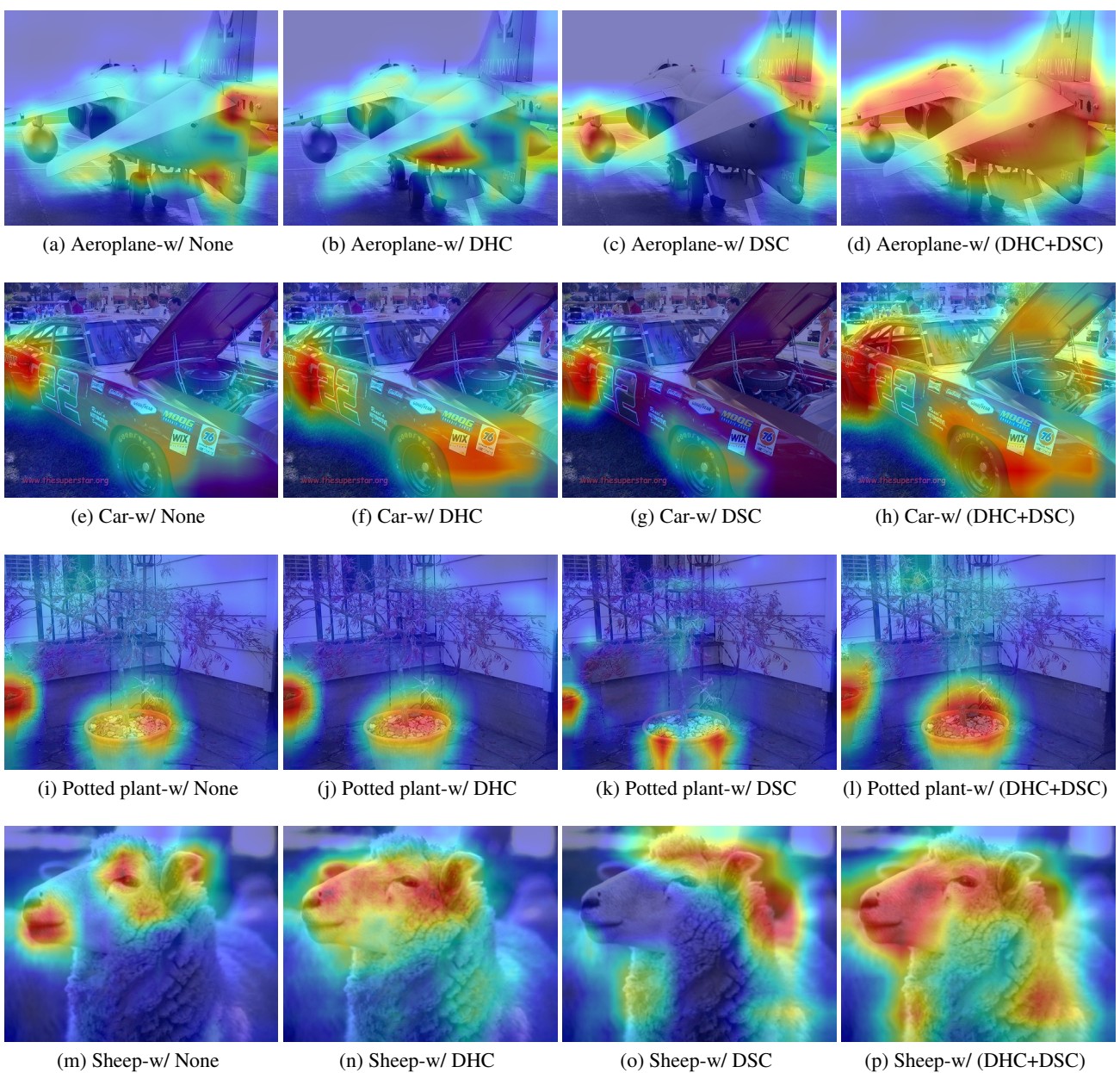

(a) Aeroplane-w/ None    (b) Aeroplane-w/ DHC    (c) Aeroplane-w/ DSC    (d) Aeroplane-w/ (DHC+DSC)

(e) Car-w/ None    (f) Car-w/ DHC    (g) Car-w/ DSC    (h) Car-w/ (DHC+DSC)

(i) Potted plant-w/ None    (j) Potted plant-w/ DHC    (k) Potted plant-w/ DSC    (l) Potted plant-w/ (DHC+DSC)

(m) Sheep-w/ None    (n) Sheep-w/ DHC    (o) Sheep-w/ DSC    (p) Sheep-w/ (DHC+DSC)

Figure 7: The visualization with different modules.

Since

$$
\begin{aligned}
\Phi_{X_{\mathcal{M}_2}} &= I(X_{\mathcal{M}_2}; Y \mid X_{\mathcal{M}_1}) \\
&= I(Y; X_{\mathcal{M}_1}, X_{\mathcal{M}_2}) - I(Y; X_{\mathcal{M}_1}) \\
&= [H(Y) - H(Y \mid X_{\mathcal{M}_1}, X_{\mathcal{M}_2})] - [H(Y) - H(Y \mid X_{\mathcal{M}_1})] \\
&= H(Y|X_{\mathcal{M}_1}) - H(Y|X_{\mathcal{M}_1}, X_{\mathcal{M}_2}).
\end{aligned}
\tag{26}
$$

We can derive

$$
\frac{H(Y|X_{\mathcal{M}_1}) - \Phi_{X_{\mathcal{M}_2}} - \log 2}{\log |Y|} \leq P_{e_c}^{mul} \leq 1 - \exp(-H(Y \mid X_{\mathcal{M}_1}) + \Phi_{X_{\mathcal{M}_2}}).
\tag{27}
$$

$\square$

### D.2. Proof of Theorem 3.5

*Proof.* We leverage previous results (Zuo et al., 2024; Li et al., 2023a) and assume that $w^*$ is the optimal parameter for the given dataset $\mathcal{D}_{\mathcal{M}_1}$. Hence, we have:

$$
\begin{aligned}
&\log p\left(w^* \mid \mathcal{D}_{\mathcal{M}_1}\right) \\
&= \log p\left(\mathcal{D}_{\mathcal{M}_1} \mid w^*\right) + \log p\left(w^*\right) - \log p(\mathcal{D}_{\mathcal{M}_1}).
\end{aligned}
\tag{28}
$$

For $p(\mathcal{D}_{\mathcal{M}_1})$, we have

$$
p(\mathcal{D}_{\mathcal{M}_1}) = \int p(\mathcal{D}_{\mathcal{M}_1} \mid w)p(w)\,dw.
\tag{29}
$$

Since

$$\int p(\mathcal{D}_{\mathcal{M}_1} \mid w)p(w)\,dw < \int p\left(\mathcal{D}_{\mathcal{M}_1} \mid w^*\right)p(w)\,dw. \quad (30)$$

We can derive

$$p(\mathcal{D}_{\mathcal{M}_1}) < p\left(\mathcal{D}_{\mathcal{M}_1} \mid w^*\right). \quad (31)$$

After integrating $\mathcal{D}_{\mathcal{M}_1}$ and $\mathcal{D}_{\Phi_{\mathcal{M}_2}}$, the combined dataset is $\mathcal{D}_S = \mathcal{D}_{\mathcal{M}_1} \cup \mathcal{D}_{\Phi_{\mathcal{M}_2}}$, yielding

$$
\begin{aligned}
&\log p(w^* \mid \mathcal{D}_S) \\
&= \log p(\mathcal{D}_S \mid w^*) + \log p(w^*) - \log p(\mathcal{D}_S) \\
&= \log p(\mathcal{D}_{\mathcal{M}_1}, \mathcal{D}_{\Phi_{\mathcal{M}_2}} \mid w^*) + \log p(w^*) \\
&\quad - \log p(\mathcal{D}_{\mathcal{M}_1}, \mathcal{D}_{\Phi_{\mathcal{M}_2}}) \\
&= \log p(\mathcal{D}_{\mathcal{M}_1} \mid w^*) + \log p(\mathcal{D}_{\Phi_{\mathcal{M}_2}} \mid w^*) + \log p(w^*) \\
&\quad - \log p(\mathcal{D}_{\mathcal{M}_1}) - \log p(\mathcal{D}_{\Phi_{\mathcal{M}_2}}) \\
&= \log p(\mathcal{D}_{\mathcal{M}_1} \mid w^*) + \log p(w^*) - \log p(\mathcal{D}_{\mathcal{M}_1}) \\
&\quad + \log p(\mathcal{D}_{\Phi_{\mathcal{M}_2}} \mid w^*) - \log p(\mathcal{D}_{\Phi_{\mathcal{M}_2}}).
\end{aligned}
\quad (32)
$$

Assuming the supplementary augmented dataset $\mathcal{D}_{\Phi_{\mathcal{M}_2}}$ closely resembles the data distribution of $\mathcal{D}_{\mathcal{M}_1}$, we have

$$p\left(\mathcal{D}_{\Phi_{\mathcal{M}_2}} \mid w^*\right) > p\left(\mathcal{D}_{\Phi_{\mathcal{M}_2}}\right). \quad (33)$$

We can derive

$$
\begin{aligned}
&\log p(w^* \mid \mathcal{D}_S) \\
&> \log p(\mathcal{D}_{\mathcal{M}_1} \mid w^*) + \log p(w^*) - \log p(\mathcal{D}_{\mathcal{M}_1}) \\
&= \log p(w^* \mid \mathcal{D}_{\mathcal{M}_1}).
\end{aligned}
\quad (34)
$$

$\square$

### D.3. Proof of Theorem 3.7

*Proof.* We decompose the DISC into two components: the backbone with parameters $w_f^*$ and the head with parameters $w_h^*$. Using the supplementary augmented dataset $\mathcal{D}_{\Phi_{\mathcal{M}_2}}$, we obtain

$$
\begin{aligned}
&\log p\left(w^* \mid \mathcal{D}_{\Phi_{\mathcal{M}_2}}\right) \\
&= \log p\left(w_f^*, w_h^* \mid \mathcal{D}_{\Phi_{\mathcal{M}_2}}\right) \\
&= \log p\left(w_f^* \mid \mathcal{D}_{\Phi_{\mathcal{M}_2}}\right) + \log p\left(w_h^* \mid \mathcal{D}_{\Phi_{\mathcal{M}_2}}\right) \\
&= \log p\left(w_f^* \mid \mathcal{D}_{\Phi_{\mathcal{M}_2}}\right) + \log p\left(w_h^* \mid f\left(\mathcal{D}_{\Phi_{\mathcal{M}_2}}\right)\right) \\
&= \log p\left(w_f^* \mid \mathcal{D}_{\Phi_{\mathcal{M}_2}}\right) + \log p\left(f\left(\mathcal{D}_{\Phi_{\mathcal{M}_2}}\right) \mid w_h^*\right) \\
&\quad + \log p\left(w_h^*\right) - \log p\left(f\left(\mathcal{D}_{\Phi_{\mathcal{M}_2}}\right)\right).
\end{aligned}
\quad (35)
$$

Since $\mathcal{D}_{\Phi_{\mathcal{M}_2}}$ is the supplementary augmentation of the dataset $\mathcal{D}_{\mathcal{M}_1}$, it has a similar distribution with $\mathcal{D}_{\mathcal{M}_1}$, we

have

$$
\begin{aligned}
&\log p\left(w_h^* \mid f\left(\mathcal{D}_{\Phi_{\mathcal{M}_2}}\right)\right) \\
&= \log p\left(f\left(\mathcal{D}_{\Phi_{\mathcal{M}_2}}\right) \mid w_h^*\right) + \log p\left(w_h^*\right) - \log p\left(f\left(\mathcal{D}_{\Phi_{\mathcal{M}_2}}\right)\right) \\
&< \log p\left(f\left(\mathcal{D}_{\mathcal{M}_1}\right)^{\Phi_{\mathcal{M}_2}} \mid w_h^*\right) + \log p\left(w_h^*\right) \\
&\quad - \log p\left(f\left(\mathcal{D}_{\mathcal{M}_1}\right)^{\Phi_{\mathcal{M}_2}}\right) \\
&= \log p\left(w_h^* \mid f(\mathcal{D}_{\mathcal{M}_1})^{\Phi_{\mathcal{M}_2}}\right).
\end{aligned}
\quad (36)
$$

We can derive

$$
\begin{aligned}
&\log p\left(w^* \mid \mathcal{D}_{\Phi_{\mathcal{M}_2}}\right) \\
&= \log p\left(w_f^* \mid \mathcal{D}_{\Phi_{\mathcal{M}_2}}\right) + \log p\left(w_h^* \mid f\left(\mathcal{D}_{\Phi_{\mathcal{M}_2}}\right)\right) \\
&< \log p\left(w_f^* \mid \mathcal{D}_{\Phi_{\mathcal{M}_2}}\right) + \log p\left(w_h^* \mid f(\mathcal{D}_{\mathcal{M}_1})^{\Phi_{\mathcal{M}_2}}\right).
\end{aligned}
\quad (37)
$$

$\square$

### D.4. Proof of Theorem 4.1

*Proof.* Let $D_{\text{train}} = \{x_i, y_i\}_{i=1}^N$ be a training dataset of $N$ samples, and $e\hat{r}r(f^m)$ be the empirical error of the $m$-th model $f^m$ on $D_{\text{train}}$. For any hypothesis $f \in \mathcal{H}$ (i.e., $\mathcal{H} : \mathcal{X} \to \{-1, 1\}$), with probability at least $1 - \delta$, the generalization error is bounded by:

$$
\begin{aligned}
\text{GError}(f) \leq &\underbrace{\sum_{m=1}^M \mathbb{E}(w^m)e\hat{r}r(f^m)}_{\text{Term-L (empirical loss)}} + \underbrace{\sum_{m=1}^M \mathbb{E}(w^m)\mathfrak{R}_m(f^m)}_{\text{Term-C (complexity)}} \\
&+ \underbrace{\sum_{m=1}^M Cov(w^m, \ell^m)}_{\text{Term-Cov (covariance)}} + M\sqrt{\frac{\ln(1/\delta)}{2N}},
\end{aligned}
\quad (38)
$$

where $\mathbb{E}(w^m)$ is the expected collaboration weights, $\mathfrak{R}_m(f^m)$ is the Rademacher complexity of model $f^m$, and $Cov(w^m, \ell^m)$ is the covariance between the weight and the loss.

In *static collaboration*, the weights $w_{\text{static}}^m$ are constant, hence

$$Cov(w_{\text{static}}^m, \ell^m) = 0. \quad (39)$$

In *dynamic collaboration*, the collaboration weight $w_{\text{dynamic}}^m$ increases as the model loss $\ell^m$ decreases. Thus,

$$Cov(w_{\text{dynamic}}^m, \ell^m) \leq 0, \quad (40)$$

which effectively reduces the Term-Cov and thereby lowers the generalization bound.

Based on the principle of convexity, it can be concluded that:

$$\sum \mathbb{E}(w_{\text{dynamic}}^m)\, e\hat{r}r(f^m) \leq \sum w_{\text{static}}^m\, e\hat{r}r(f^m). \quad (41)$$

$$\sum \mathbb{E}(w_{\text{dynamic}}^m)\, \mathfrak{R}_m(f^m) \leq \sum w_{\text{static}}^m\, \mathfrak{R}_m(f^m). \quad (42)$$

Since the confidence term $M\sqrt{\frac{\ln(1/\delta)}{2N}}$ is independent of the collaboration strategy, it remains the same. Therefore, suppose the hypothesis space is $\mathcal{H}: \mathcal{X} \to \{-1, 1\}$. Then for any $f_{\text{dynamic}}, f_{\text{static}} \in \mathcal{H}$, and for $1 > \delta > 0$, it holds that

$$\mathcal{O}(\text{GError}(f_{\text{dynamic}})) \leq \mathcal{O}(\text{GError}(f_{\text{static}})). \quad (43)$$

$\square$

## E. Future Applications and Future Research

In this section, we provide an in-depth exploration of SC's insights across diverse real-world applications and learning paradigms, highlighting its potential to bridge disciplinary boundaries. By shedding light on its foundational principles and practical implications, we aim to spark broader interest in this rapidly evolving and interdisciplinary domain.

### E.1. Future Applications of SC

This section delves into the future applications of SC in emerging fields. We have studied various potential application scenarios, analyzed their unique requirements, challenges, and opportunities, explored the different uses of SC across different domains, and provided practical guidance on how to implement it.

**Defect detection:** Defect detection and segmentation tasks for industrial products typically involve processing high-dimensional, noisy image data. Moreover, the manifestations of faults are diverse, involving different materials, structures, and processes, and may appear as small cracks, corrosion, wear, etc. This requires models to handle complex and heterogeneous data in a diversified manner, which poses a significant challenge for the application of SC. Future research should focus on ensuring that the coevolution between different models effectively enhances their individual performance without causing a decline in performance. Additionally, the interaction between detection and segmentation models requires precise design, enabling them to mutually promote each other's learning, rather than simply optimizing their own objectives independently.

**Autonomous driving:** In autonomous driving, road environment segmentation and object detection tasks need to handle complex and dynamic scenarios, including varying weather conditions, visibility, and road situations. Ensuring that SC between different models can effectively enhance each other's performance in such dynamically changing environments is a significant challenge. By applying SC to autonomous driving, the detection model can assist the segmentation model in accurately identifying different traffic signs, pedestrians, vehicles, and other objects, while the segmentation model can provide more precise background information and spatial distribution for object detection. Future research should focus on designing adaptive feedback mechanisms that allow models to mutually promote each other during joint training and flexibly adjust according to the demands of different scenarios and tasks.

**Emergency rescue:** In emergency rescue scenarios, Augmented Reality (AR) technology can play a crucial role. In complex rescue environments, AR can help rescue personnel quickly find the best route by real-time planning and displaying safe paths and obstacle avoidance information. For lost item classification, AR can overlay real-time classification data and item features onto the real-world scene, assisting rescue personnel in quickly identifying and categorizing items. The application of SC in AR for emergency rescue scenarios presents the following challenges. Emergency rescue scenes are typically highly dynamic and uncertain, requiring models to not only have high accuracy in detection and classification but also to possess strong robustness. Additionally, real-time detection and classification tasks have different requirements: object detection focuses on quickly and accurately locating and identifying key targets at the rescue site, while item classification focuses on the rapid classification and archiving of items. In this context, research should focus on designing SC mechanisms that enable the two models to mutually enhance each other, creating a synergistic effect.

### E.2. Future Research and Expansion of SC

In addition to the future applications of SC mentioned above, in future research, SC can also be integrated with various existing learning paradigms to be further extended, achieving better performance across a range of tasks.

**Federated Learning (FL):** FL is a form of distributed learning that allows models on different devices to be trained locally and only share updates with a central server. This approach ensures data privacy, making it particularly suitable for collaborative learning in multi-device environments. Future research should focus on ensuring collaborative learning between models, optimizing communication efficiency, and accelerating the evolution process in multi-device environments, while adhering to data privacy protection requirements.

In the context of FL, models on different devices can autonomously evolve locally and share the results of their

local updates through a socialization mechanism to enhance collaborative effects. For example, in industrial defect detection tasks, due to the involvement of proprietary research and development secrets in industrial products, full open-source sharing is not feasible. However, companies can contribute their defect detection models, which can then be shared via the socialization mechanism on a central server, driving advancements in industrial defect detection and segmentation. The integration of SC with FL enables models distributed across different locations to collaborate in evolution, safeguarding data privacy while enhancing the overall intelligence level of the system.

**Continual Learning (CL):** CL aims to enable models to gradually expand their knowledge while continuously receiving new tasks and data, without forgetting previously learned information. This learning paradigm is particularly suitable for intelligent systems that operate over long periods. Future research should focus on combining continual learning with SC, ensuring that models can share data and knowledge while avoiding catastrophic forgetting.

In the framework of CL, models can autonomously evolve when faced with constantly changing environments, and share local updates through a socialization mechanism to enhance collaborative effects and reduce forgetting. For example, in autonomous driving tasks, as road environments and traffic conditions continuously change, new objects or scenarios may emerge, such as suddenly appearing wild animals, different forms of accident vehicles, or even unseen obstacles. Although these new targets or situations might represent entirely new categories, through CL, the autonomous driving system can gradually adapt and expand its knowledge base, while SC helps prevent catastrophic forgetting, ensuring that the model does not lose the ability to handle previous tasks while adapting to new ones. The integration of SC and CL enables autonomous driving models in different environments to evolve collaboratively, not only enhancing model adaptability but also ensuring the system as a whole maintains stronger robustness and performance over the long term.

**Online Learning (OL):** OL is a process in which models continuously learn from streaming data, typically updating as data arrives gradually, thus avoiding the computational burden associated with batch training. SC can leverage the real-time feedback mechanism of OL, allowing different models to update collaboratively based on the latest data, rather than waiting for batch data accumulation.

In a constantly changing environment, the coevolution between models can accelerate their learning process and ensure that collaborative optimization does not cause performance fluctuations due to the dynamic changes in the data stream. For example, in emergency rescue scenarios, due to the complexity of on-site conditions, the collected data often contains missing values, noise, and biases, leading to potential shifts in sample distribution. This presents a major challenge for OL. However, within the SC framework, different models can collaborate with each other, where the detection model can help the segmentation model identify missing or partially damaged objects, and the segmentation model can provide more precise contextual information to improve the detection model's adaptation to the environment. In this way, SC can enhance the model's adaptability, ensuring that it maintains high performance and robustness even when faced with distribution shifts and drift in the data.

