# OpenReview forum: "Socialized Coevolution: Advancing a Better World through Cross-Task Collaboration"
_ICML.cc/2025/Conference — ICML 2025 poster_

### Official Review · Reviewer_LUex · 2025-03-09

**Overall Recommendation:** 3

**Summary:**

This paper introduces a practical learning paradigm of Socialized Coevolution (SC), to overcome the limitations of methods that rely heavily on data-driven, single-model approaches and often fall short in leveraging inter-model knowledge transfer. The paper argues that existing methods like Knowledge Distillation (KD) and Multi-Task Learning (MTL) suffer from issues such as blind imitation, loss of task-specific expertise, and a lack of dynamic interaction. The learning paradigm is the Dynamic Information Socialized Collaboration (DISC) framework. DISC incorporates two key modules: Dynamic Hierarchical Collaboration (DHC) for feature-level interaction and Dynamic Selective Collaboration (DSC) for logit-level interaction.

**Claims And Evidence:**

While the paper presents experimental results that show performance improvements using DISC, I would suggest that some claims could use some further support. For instance, the claim that DISC 'balances specialization and generalization' is somewhat weakly substantiated. The experiments do demonstrate improved average performance and ablation studies show that DHC and DSC are beneficial individually, and even more so together, but they don't deeply analyze how these modules specifically achieve this balance. Is it truly about specialization and generalization, or is it, for example, primarily driven by improved representation learning?

Furthermore, the theoretical analysis, while interesting, does feel somewhat disconnected from the empirical results. Theorems 3.5-3.8 are presented, but the paper lacks a clear explanation of how these theoretical findings/properties directly predict or explain the observed performance gains of DISC. More explicit connections between the theory and what these imply for the empirical results could be useful for improving the claims. Finally, the titles's broad claim of "Advancing a Better World" is hyperbolic and not necessarily supported by the paper's scope. This statement should be replaced with a more grounded description of the paper's technical contributions and their potential impact within the machine learning community.

**Essential References Not Discussed:**

NA

**Experimental Designs Or Analyses:**

The experiment design is generally sound.

The ablation study (Table 3) is a good starting point, but it could be more comprehensive. While it ablates DHC and DSC individually, it would also be informative to see ablations of specific components within DHC and DSC to understand the contribution of each sub-module. For instance, where any other forms tested for the progressive interaction mode in DHC? What if the auxiliary information is not integrated gradually? In DSC, what if we don't use the adaptive weighting?

The paper reports mean performance values, but should also report standard deviations or confidence intervals, especially for the results that are only slight improvements.

**Methods And Evaluation Criteria:**

The proposed DISC method appears to be a reasonable approach to SC. The design of DHC and DSC modules, focusing on hierarchical feature interaction and dynamic selective logit collaboration, is logically motivated by both the identified limitations of existing KD and MTL methods, and Sec 4.1. Regarding baselines, the selection is relatively broad, including KD and MTL variants, as well as self-supervised methods.

**Other Comments Or Suggestions:**

NA

**Other Strengths And Weaknesses:**

**Strengths**
- Novelty - The paper introduces a genuinely novel learning paradigm. Drawing inspiration from human social learning and translating it into a practical algorithmic framework is a fresh perspective on knowledge transfer and collaborative learning in AI. The specific design of DISC, with its DHC and DSC modules for dynamic hierarchical and selective interaction, further contributes to the originality of the approach.
- **Clarity:** The paper is generally well-written and structured, with clear explanations of the motivation, methodology, and experimental setup. The figures and tables are mostly informative and contribute to understanding the approach.

**Weaknesses**
- **Scope of Evaluation:** The evaluation, while using standard benchmarks, might be limited in fully showcasing the potential of SC in more complex and real-world scenarios. Demonstrating SC's effectiveness on more diverse, challenging, and larger-scale datasets would significantly strengthen the empirical validation.
- **Scalability and Complexity:** The paper does not explicitly address the scalability of SC. It's unclear if the dynamic interaction mechanisms in DISC will remain efficient and effective as the "society" of models grows. Is the society of models limited to two models? Furthermore, the computational complexity of DISC compared to simpler KD or MTL methods should be discussed.
- **Limitations**: No obvious discussion of the limitations of the framework.

**Questions For Authors:**

1. The authors propose some interesting future applications of the work in the appendix, however I do not know what the difficulties are for scaling to this problems? How well does this framework scale to any of the problems discussed as future applications in the appendix?

**Relation To Broader Scientific Literature:**

Overall, the paper makes a valuable contribution and extends existing research in Knowledge Distillation (KD) and Multi-Task Learning (MTL). The paper's unique aspect is drawing inspiration from Social Learning (SL) in human societies and cognitive science and it is well positioned at the intersection of these fields.

**Theoretical Claims:**

I have not checked the correctness of the proofs, and would defer to a reviewer with a stronger expertise in information theory to do so.

---

> ### Author Rebuttal · Authors · 2025-03-31
>
> Thank you for your thoughtful feedback. We provide detailed responses to the issues. Please refer to the anonymous link (https://anonymous.4open.science/r/Supplementary-materials-for-SC/) for the supplementary tables and proofs.
>
> >Q1. Specialization-generalization trade-off
>
> R1. DISC balances specialization and generalization by combining interactive learning with a dynamic master–assistant architecture, enhancing individual capabilities and task-level collaboration.
>
> - **Dynamic master–assistant architecture enables balance:** For each task, DISC adaptively assigns the model most suited to the task as the master, with others serving as assistants. This role assignment ensures specialization without sacrificing generalization.
>
> - **Interactive learning enhances specialization:** Classification (CLS) emphasizes image-wise, while detection (DET) attends to patch-wise. Specialization is enhanced through interaction.
>
> >Q2. Connection between theory and empirical results
>
> R2. We further clarify the connection between our theory and DISC through additional experiments (Table 9) and analysis. Sociability Information (SI) exists in both data and models (Definition 3.2), and Theorems 3.5 and 3.7 suggest that enhanced data and model representations enable SC. Therefore, we observed performance under different data and model enhancements, finding that DISC can better leverage SI (Theorem 3.3).
>
> - **Theory-guided method:** 1) SI (Definition 3.2) is the premise of SC (Theorem 3.3). 2) Data augmentation improves SC (Theorem 3.5). 3) Representation enhancement improves SC (Theorem 3.7). 4) Driven by SI, we enhance the dataset through task-sharing and design DISC with DHC and DSC modules, improving representational capacity and achieving SC.
>
> - **Theory-empirical relationship:** In machine society, data represents direct experience, and model interaction reflects indirect experience. Theorem 3.5 shows that diverse datasets strengthen the direct experience, while Theorem 3.7 highlights that expert model interaction improves the indirect experience. DISC leverages both through shared datasets and enhances representations via DHC and DSC, enabling SC (Table 9).
>
> >Q3. Paper title
>
> R3. If allowed by the conference, we will revise the title to “Socialized Coevolution: Cross-task Collaboration Driven by Dynamic Interactions”, which better reflects the technical contribution.
>
> >Q4. Ablation on progressive interaction and adaptive weighting
>
> R4. Progressive interaction and adaptive weighting allow models to dynamically collaborate per sample without overwhelming the main model. In contrast, direct or static strategies often lead to excessive interaction and degrade learning.
>
> - **Progressive interaction:** We explored sine- and cosine-based progressive modes under varying thresholds (Fig. 8 in Appendix). Sudden or deep interactions early in training can disrupt the main model, while gradual interaction maintains learning stability.
>
> - **Direct integration:** We tested non-progressive integration by setting auxiliary weights to 1 (Fig. 3 in paper and Fig. 7 in Appendix). Results show that abrupt inclusion of auxiliary information fails to improve representation and may even hinder learning.
>
> - **Static weighting:** If an optimal combination existed, only one red patch would appear (Fig. 3 in paper). The multiple red patches motivate dynamic weighting to enhance inter-model interactions. We provide a theoretical analysis showing that dynamic weighting outperforms static weighting [1], with dynamic weight negatively correlated with individual loss and positively correlated with collaborator loss (Eq. 19).
>
> $$
> \begin{align}
> &GE(f)\\leq |\\mathcal{M}| \\left( \\mathcal{R}_N(\\mathcal{H}) + \\sqrt{\\frac{\\ln(1/\\Delta)}{2N}} \\right)+\\sum\_{m=1}^{|\\mathcal{M}|} \\hat{err}(f^m) \\ + \\sum\_{m=1}^{|\\mathcal{M}|} \\left[ \\frac{1}{|\\mathcal{M}|} \\underbrace{Cov(\\omega^m, \\ell^m)}\_{\\textit{Negative self-correlation in DSC}} - \\frac{|\\mathcal{M}|-1}{|\\mathcal{M}|} \\sum\_{j \\neq m} \\underbrace{Cov(\\omega^m, \\ell^j)}\_{\\textit{Positive collaborative correlation in DSC}} \\right]
> \end{align}
> $$
>
> [1] Provable dynamic fusion for low-quality multimodal data. ICML, 2023.
>
> >Q5. Evaluation on diverse and large-scale datasets
>
> R5. We additionally conducted experiments (including standard deviations) on CIFAR10, Food-101, and WIDER FACE (Table 10). Consistent results across multiple datasets show that DISC achieves better performance through dynamic interaction and collaboration.
>
> >Q6. Scalability and complexity
>
> R6. With sufficient computational power, DISC can scale to any number of ensembles. For pairwise interactions, the total cost for N models is N(N−1)K, where K represents the per-interaction cost. KD methods are efficient but sacrifice generalization for specialization (Table 11). Compared to MTL, DISC incurs lower costs while better balancing specialization and generalization, facilitating other paradigms and applications.

---

### Official Review · Reviewer_7Pyh · 2025-03-11

**Overall Recommendation:** 3

**Summary:**

This paper proposes a practical learning framework, DISC, to improve model performance across two task domains. The proposed method allows two expert models to exchange intermediate states and weights in a structured manner to achieve superior performance in both task domains. The authors evaluated DISC on image classification and detection tasks, comparing it to SOTA methods in fine-tuning, knowledge distillation, and multi-task learning. Results show that DISC outperforms baselines in both original and transfer tasks.

**Claims And Evidence:**

I am not totally convinced by the definition of socialized coevolution where the authors metaphorize two models exchanging intermediate outputs as "collaboration" and exchanging weights as "communication." To me, the proposed method is more of a practical learning framework than a social learning process where autonomous agents interact and learn with each other. More empirical evidence is needed to justify the connection between the proposed methods and social learning. For example, does DISC have any similar characteristics to social learning?

As claimed to be a unified learning framework, social coevolution needs to be evaluated in diverse domains and model structures. Given the social motivation, it would be interesting to extend the proposed method from two models to multiple models.

The hierarchical and progressive parts of DISC are based on human prior knowledge (i.e., only exchanging adjacent layers and gradually introducing auxiliary information) instead of learning from data. Does this impact the generalization of the proposed method to different model structures and task domains?

**Essential References Not Discussed:**

No

**Experimental Designs Or Analyses:**

It is unclear to me how Figure 3 is generated and how this visualization led to the dynamic weighting mechanism.

**Methods And Evaluation Criteria:**

The experiment design and evaluation process looks convincing to me.

**Other Comments Or Suggestions:**

There is a minor format issue that the paper has a header of Submission and Formatting Instructions for ICML 2025.

**Other Strengths And Weaknesses:**

No

**Questions For Authors:**

No.

**Relation To Broader Scientific Literature:**

No

**Theoretical Claims:**

In the proof of Theorems 3.5 and 3.7, the assumption that the supplementary augmented dataset DM2 closely resembles the data distribution of DM1 might not hold in situations where the two tasks/domains are loosely connected. It is less clear how the nature of multiple objective tasks and the distribution of datasets influence the effectiveness of social co-evolution.

---

> ### Author Rebuttal · Authors · 2025-03-31
>
> Thank you for your valuable feedback. We provide detailed responses to the issues. For the supplementary tables and proofs, please refer to the anonymous link (https://anonymous.4open.science/r/Supplementary-materials-for-SC/).
>
> >Q1. Connection with social learning
>
> R1. Human and machine societies share a common principle: individuals improve through interaction and collectives through collaboration [1-2]. Both DISC and Social Learning (SL) [3] emphasize acquiring knowledge directly and indirectly, which is empirically validated (Table 9). DISC is inspired by social learning (Eq. 16): its layer-wise hierarchy (Eq. 17), DHC (Eq. 18), and DSC (Eq. 19) correspond to social hierarchy, progressive learning, and cultural learning (Table 7), respectively.
>
> - **SL inspires DISC in machine societies.** SL involves learning through observation (direct experience) and interaction (indirect experience), while DISC enables Socialized Coevolution (SC) through both data (direct experience)  and models (indirect experience), reflecting SL's dual nature.
>
> - **Empirical evidence:** Experiments (Table 9) show that direct learning from data builds basic ability, while indirect learning from collaborators depends on their ability. DISC combines both to realize SC through SL.
>
> [1] A social path to human-like artificial intelligence. NMI, 2023.
>
> [2] The secret of our success: How culture is driving human evolution, domesticating our species, and making us smarter. Princeton University Press, 2016.
>
> [3] Social learning theory. Prentice Hall, 1977.
>
> >Q2. Scalability and generalization across models and tasks
>
> R2.   We analyze scalability and generalization from the backbone, training strategy, and task (Table 8), and find that sociability benefits from expert-based models.
>
> - **Backbone:** ResNet152 outperforms ResNet50 due to its design and stronger representational capacity.
>
> - **Training strategy:** Among ResNet50 variants, ImageNet pre-training yields the best performance, benefiting from the scale and diversity of the ImageNet dataset.
>
> - **Task:** Each task corresponds to a different model. Classification (CLS), detection (DET), and segmentation (SEG) focus on image-, patch-, and pixel-wise, respectively, exhibiting sociability.
>
> >Q3. Influence of tasks and datasets characteristics on SC
>
> R3. Tasks and data with higher sociability information (SI) better support SC (Theorem 3.3). As in multi-view learning [1], consistency and complementarity among views are essential for effective representation, SC similarly relies on SI (Definition 3.2) among models. Theorem 3.3 proves that SI is a prerequisite for SC and positively correlated with it.
>
> - **SI-SC positive correlation:** In Definition 3.2, we formalize SI as $$\\Phi\_{\\mathcal{M}\_2} = I(X\_{\\mathcal{M}\_2}; Y \\mid X\_{\\mathcal{M}\_1})$$ Bayes error rate shows a positive correlation between SI (e.g., tasks and datasets) and SC.
>
> $$
> \begin{align}
>     \\frac{{H}({Y}|{X}\_{\\mathcal{M}\_{1}})-\\Phi\_{\\mathcal{M}\_{2}}-\log 2}{\log |{Y}|}
>             \\leq {P}^{mul}\_{e\_c} \\leq 1-\\exp (-{H}({Y}\\mid {X}\_{\\mathcal{M}\_{1}})+\\Phi\_{\\mathcal{M}\_{2}})
> \end{align}
> $$
>
> - **Data-level SI arises from diverse samples (Theorem 3.5):** Structurally diverse datasets, like CIFAR-100 (more classes but low resolution) and VOC07+12 (fewer classes with high resolution and rich scene complexity), offer complementary information, enhancing SI and improving SC.
>
> - **Task-level SI derives from multi-wise information (Theorem 3.7):** While all tasks aim to learn better representations, they focus on different levels, CLS (image-wise), DET (patch-wise), and SEG (pixel-wise). This complementarity yields SI, which benefits SC.
>
> [1] A survey of multi-view representation learning. IEEE TKDE, 2018.
>
> >Q4. Interpretation of Fig.3
>
> R4. Fig. 3 shows performance with fixed weight combinations for features and logits. If an optimal combination existed, only one red patch would appear. The multiple red patches motivate dynamic weighting to enhance inter-model interactions. We provide a theoretical analysis demonstrating that dynamic weighting outperforms static weighting [1]. Specifically, the dynamic weight is negatively correlated with individual loss and positively correlated with collaborator loss (Eq. 19).
>
> $$
> \begin{align}
> &GE(f)\\leq |\\mathcal{M}| \\left( \\mathcal{R}_N(\\mathcal{H}) + \\sqrt{\\frac{\\ln(1/\\Delta)}{2N}} \\right)+\\sum\_{m=1}^{|\\mathcal{M}|} \\hat{err}(f^m) \\ + \\sum\_{m=1}^{|\\mathcal{M}|} \\left[ \\frac{1}{|\\mathcal{M}|} \\underbrace{Cov(\\omega^m, \\ell^m)}\_{\\textit{Negative self-correlation in DSC}} - \\frac{|\\mathcal{M}|-1}{|\\mathcal{M}|} \\sum\_{j \\neq m} \\underbrace{Cov(\\omega^m, \\ell^j)}\_{\\textit{Positive collaborative correlation in DSC}} \\right]
> \end{align}
> $$
>
> [1] Provable dynamic fusion for low-quality multimodal data. ICML, 2023.
>
> >Q5. Formatting issue
>
> R5. Thank you for your suggestions, we will fix it in the final version.

---

### Official Review · Reviewer_aRWf · 2025-03-13

**Overall Recommendation:** 3

**Summary:**

The authors introduce a learning paradigm called Socialized Coevolution (SC), designed to enhance the performance of existing tasks with the support of an auxiliary model. Specifically, the main model progressively integrates hierarchical auxiliary information through a dynamically weighted communication mechanism. Experimental results on the CIFAR-100 and VOC07+12 datasets demonstrate the effectiveness of the proposed method compared to baseline approaches.

**Claims And Evidence:**

Yes.

**Essential References Not Discussed:**

The authors mention that the proposed method is inspired by cultural evolution; however, a more detailed discussion on related previous works and their connection to this approach would be beneficial. Additionally, further clarification on the differences and similarities between the proposed setting and model merging would help provide a clearer understanding.

**Experimental Designs Or Analyses:**

The authors have conducted corresponding tests on multiple tasks (classification and detection) and provided ablation experiments to demonstrate the effectiveness of the proposed method.

The comparison between the proposed method and KD/MTL in Table 1 raises some concerns. For example, it is unclear why the MTL setting involves freezing the model rather than training it with a specific method. Additionally, further analysis is provided.

**Methods And Evaluation Criteria:**

The proposed method is evaluated on CIFAR-100 for the classification task and VOC07+12 for the detection task, using only 10% of the training set. The comparisons are conducted against classical methods under the settings of Knowledge Distillation (KD) and Multi-Task Learning (MTL).

**Other Comments Or Suggestions:**

All suggestions are included in the "Strengths and Weaknesses" section.

**Other Strengths And Weaknesses:**

Strengths:
The paper shows the possibilities that the hierarchical information can complementary for each other, as the combination of them shows improvements.

Weaknesses:

1.Although the authors claim to analyze the proposed method from an information-theoretic perspective, the connection between the theoretical framework and the proposed method appears to be weak.

2.The proposed method lacks a solid theoretical foundation. For example, the specific motivations for employing the sine function in the progressive interaction module and the rationale behind the dynamic weighting mechanism are not adequately discussed.

3.The training procedure is not clearly described. In the supplementary material, the pseudo-algorithm presents the joint training of different tasks. However, critical details regarding the training process, such as the number of iterations and other relevant training parameters, are not provided. Furthermore, it remains unclear whether the proposed DSC module is employed during inference.

4.The robustness and the generalization of the proposed method has not been thoroughly analyzed. Specifically, it is unclear whether the method performs consistently across different backbone architectures, including both pre-trained backbones and those trained from scratch, as well as different backbones across different tasks. Additionally, the generalizability of the proposed method to a broader range of tasks has not been sufficiently examined.

**Questions For Authors:**

All questions are included in the "Strengths and Weaknesses" section.

**Relation To Broader Scientific Literature:**

The proposed problem is closely related with prior research in relevant areas, such as knowledge distillation (KD), multi-task learning (MTL), continual learning (CL) and federated learning (FL).

**Theoretical Claims:**

The authors present theoretical analysis about the sociability Information.

---

> ### Author Rebuttal · Authors · 2025-03-30
>
> Thank you for your insightful feedback. We provide detailed responses to the issues. Please refer to the link (https://anonymous.4open.science/r/Supplementary-materials-for-SC/) for the Additional tables and proofs.
>
> >Q1. MTL setting
>
> R1. The MTL setting is introduced to validate Sociability Information (SI) as the premise of Socialized Coevolution (SC). We include MAD [1] and Pix2SeqV2 [2] as MTL baselines (Table 6), emphasizing the specialization-generalization trade-off in SC.
>
> - **SI as the premise of SC:** Freezing the backbone reveals common capacity. Classification (CLS) and detection (DET) focus on image- and patch-wise, respectively, and exhibit the potential of SC (Eq. 16) due to their SI (Definition 3.2).
>
> - **Characteristics of MTL:** MAD and Pix2SeqV2 show that enhancing generalization sacrifices specialization.
>
> - **Trade-off in MTL:** MAD and Pix2SeqV2 enhance generalization at the cost of specialization, while freezing the backbone preserves specialization but limits generalization.
>
> [1] Masked AutoDecoder is effective multi-task vision generalist. CVPR, 2024.
>
> [2] A unified sequence interface for vision tasks. NeurIPS, 2022.
>
> >Q2. Relation to cultural evolution
>
> R2. Human and machine societies share a common principle: individuals improve through interaction and collectives through collaboration [1-2]. Cultural evolution includes Social Hierarchy (**SH**), Progressive Learning (**PL**), and Cultural Learning (**CL**), which motivates the SC (Eq. 16) and DISC (Table 7).
>
> - **SH:** Rooted in human societies, we use layer-wise hierarchies to enable structured collaboration (Eq. 17).
>
> - **PL:** Inspired by PL in human education, we use progressive interaction modes to bridge capacity gaps across models (Eq. 18).
>
> - **CL:** CL mirrors guided learning in human culture, informing our strong-guided communication mechanisms with capability awareness (Eq. 19).
>
> [1] A social path to human-like artificial intelligence. NMI, 2023.
>
> [2] The secret of our success: How culture is driving human evolution, domesticating our species, and making us smarter. Princeton University Press, 2016.
>
> >Q3. Differences and similarities from model merging
>
> R3. Model merging and SC share similar knowledge sources, but they differ in premises and goals.
>
> - **Differences:** 1) *Premises*. Model merging assumes redundancy in parameters and orthogonality of task vectors, whereas SC is grounded in sociability. 2) *Goals*. Model merging aims to get a better-merged model, overlooking individual evolution. In contrast, SC improves individual models via interaction and the collective via collaboration, achieving coevolution at both levels.
>
> - **Similarities:** Both acquire knowledge from other models.
>
> >Q4. Connection between theory and methodology & theoretical foundation
>
> R4. We theoretically prove stronger SI leads to better SC (Theorem 3.3). In dynamic machine societies, SI manifests through data and models, enhancing their capacity to facilitate SC (Theorems 3.5 and 3.7). Thus, we propose the dynamically SI-driven DISC and further provide a theory.
>
> - **Theory-guided method:** 1) Definition 3.2 formalizes SI, and Theorem 3.3 proves SI is essential to SC. 2) Theorems 3.5 and 3.7 show SI exists in both data and models. 3) We propose DISC based on SC (Theorem 3.3), and design DHC and DSC from different tasks (Theorems 3.5 and 3.7), offering insights into SI utilization (Eq. 20).
>
> - **Theory based on dynamic collaboration:** Dynamic outperforms static [1], with dynamism manifested through progressive interactions and dynamically weighted collaboration. Weights of DSC (Eq. 19) negatively correlate with individual loss and positively correlate with collaborator loss.
>
> $$
> \begin{align}
> &GE(f)\\leq |\\mathcal{M}| \\left( \\mathcal{R}_N(\\mathcal{H}) + \\sqrt{\\frac{\\ln(1/\\Delta)}{2N}} \\right)+\\sum\_{m=1}^{|\\mathcal{M}|} \\hat{err}(f^m) \\ + \\sum\_{m=1}^{|\\mathcal{M}|} \\left[ \\frac{1}{|\\mathcal{M}|} \\underbrace{Cov(\\omega^m, \\ell^m)}\_{\\textit{Negative self-correlation in DSC}} - \\frac{|\\mathcal{M}|-1}{|\\mathcal{M}|} \\sum\_{j \\neq m} \\underbrace{Cov(\\omega^m, \\ell^j)}\_{\\textit{Positive collaborative correlation in DSC}} \\right]
> \end{align}
> $$
>
> [1] Provable dynamic fusion for low-quality multimodal data. ICML, 2023.
>
> >Q5. Training and inference details
>
> R5. For CLS and DET, we set 79 and 147 iterations per epoch, respectively. DSC is active in training and inference. For more details, please see Section B in Appendix.
>
> >Q6. Robustness and generalization
>
> R6. We assess robustness and generalization across the backbone, training strategy, and task (Table 8), finding sociability benefits from expert-based models.
>
> - **Backbone:** ResNet152 outperforms ResNet50 due to its design and better capacity.
>
> - **Training strategy:** ImageNet pre-training yields the best result, benefiting from the scale of the data.
>
> - **Task:** CLS, DET, and SEG focus on image-, patch-, and pixel-wise, respectively, exhibiting sociability.

---

> > ### Comment · Reviewer_aRWf · 2025-04-05
> >
> > Some of my concerns are addressed. I have read the reviews and response  of other reviewer. I would like to increase my score to 3.

---

### Decision · Program_Chairs · 2025-05-01

**Decision:**

Accept (poster)

**Comment:**

The reviewers broadly agree that this paper should be accepted, noting its novel and well-motivated concept of socialized coevolution, the practical framework it introduces for investigating socialized coevolution, that its contributions draw from multiple areas (e.g., distillation, MTL, and cognitive-inspired AI), and promising results on standard benchmarks. There were some concerns with the theoretical grounding of socialized coevolution (which is somewhat disconnected from the empirical results), scalability, and some of the methodological choices — these issues were mostly clarified during the rebuttal. Consequently, the authors are strongly urged to address these concerns and incorporate aspects of the discussion into the final version of this paper. Despite these few, relatively minor concerns, the paper seems like it overall is a solid contribution.